

# Generalized hydrodynamics of classical integrable field theory: the sinh-Gordon model

Alvise Bastianello[1], Benjamin Doyon[2], Gérard Watts[2] and Takato Yoshimura[2]

**1** SISSA & INFN, via Bonomea 265, 34136 Trieste, Italy
**2** Department of Mathematics, King's College London, Strand, London WC2R 2LS, U.K.

## Abstract

Using generalized hydrodynamics (GHD), we develop the Euler hydrodynamics of classical integrable field theory. Classical field GHD is based on a known formalism for Gibbs ensembles of classical fields, that resembles the thermodynamic Bethe ansatz of quantum models, which we extend to generalized Gibbs ensembles (GGEs). In general, GHD must take into account both solitonic and radiative modes of classical fields. We observe that the quasi-particle formulation of GHD remains valid for radiative modes, even though these do not display particle-like properties in their precise dynamics. We point out that because of a UV catastrophe similar to that of black body radiation, radiative modes suffer from divergences that restrict the set of finite-average observables; this set is larger for GGEs with higher conserved charges. We concentrate on the sinh-Gordon model, which only has radiative modes, and study transport in the domain-wall initial problem as well as Euler-scale correlations in GGEs. We confirm a variety of exact GHD predictions, including those coming from hydrodynamic projection theory, by comparing with Metropolis numerical evaluations.



# 1 Introduction

Integrability has provided the backbone for a wide array of exact results in theoretical physics, in as diverse frameworks as quantum chains and classical fields. Despite its development over many years, however, until recently it remained unknown how to access dynamical quantities out of equilibrium efficiently. What happens to integrable many-body systems in situations where homogeneity and stationarity are broken, and where non-trivial currents exist? This problem has become of high importance in particular in the quantum realm, thanks to the advent of cold-atom experiments [1–5].

The lack of homogeneity breaks most of the standard structures at the core of many-body integrability, such as the inverse scattering method. Nevertheless, these standard structures may be used in conjunction with the idea of emergence of hydrodynamics. With weak, large-scale inhomogeneity and non-stationarity, one describes a many-body state in terms of fluid cells: mesoscopic regions which are assumed homogeneous, stationary and very large compared to microscopic scales, where entropy is maximized.

The limit where this description becomes exact is often referred to as the "Euler scale". This is the scaling limit whereby parameters characterizing the state are taken to vary in space on an infinitely large scale, observables are at space-time points growing with this scale and appropriately averaged over fluid cells, and correlations are likewise scaled. See for instance [6]. Various microscopic distances are expected to control the approximate validity of the limiting behaviour in real situations, including the inter-particle distance and the scattering length; however there is, as of yet, no general and precise theoretical understanding for the emergence of hydrodynamics in deterministic systems.

The corresponding Euler-scale dynamics, the dynamics on the scaled space-time, is simply

a consequence of microscopic conservation laws, and translates to a dynamics for the Lagrange parameters of each fluid cell. Within fluid cells, the usual techniques of integrability can be applied. The theory that implements these ideas in the context of quantum integrability is generalized hydrodynamics (GHD) [7,8] (see also [9–26] for recent developments and applications), a hydrodynamic theory based on the observation that in quantum integrable systems, entropy maximization gives rise to generalized Gibbs ensembles (GGEs) [27–30]. See [18,25] for a description of the Euler scaling limit in the integrability context.

The goal of this paper is to extend these ideas to integrable *classical* field theory. Focussing on the classical sinh-Gordon model, we confirm that GHD can indeed be applied to classical fields, thus further suggesting its large universality. We explicitly compare GHD predictions with Metropolis simulations of classically fluctuating initial states evolved deterministically with the field's equations of motion. We study both the partitioning protocol (or domain wall initial condition) [31–37] (see also [7, 8, 12–14, 19, 24, 38], where two halves are initialized in different homogeneous GGEs and suddenly joined at one point, and Euler-scale dynamical correlation functions in homogeneous thermal states. Our study provides in particular the first numerical tests for the recent GHD constructions of Euler-scale correlation functions in integrable models [18, 25]. In particular, we provide numerical evidence for the necessity for fluid-cell averaging in the Euler scaling limit of correlation functions.

The most powerful formulation of GHD is obtained in the quasi-particle picture. This picture is natural from Bethe ansatz integrability of quantum systems, where quasi-particles relate to Bethe roots: in this formulation, GHD is based on the formalism of the thermodynamic Bethe ansatz (TBA) [39–41]. The resulting GHD equations have also been observed to apply to certain classical gases such as the hard-rod gas [6, 13, 42, 43] or soliton gases [44–47]. In these cases, the quasi-particles represent the solitons themselves, or any dynamical objects with equivalent scattering features [17]. The general hydrodynamic principles behind GHD should, however, be applicable somewhat more generally, independently from an *a priori* underlying quasi-particle concept, such as in integrable field theory.

The theory of Gibbs states of classical integrable fields has already been developed in [48–51], and is easy to extend to GGEs. Interestingly, it takes a very similar form to that of the quantum TBA. The exact form of GGEs, and therefore of GHD, depends on the precise type of modes admitted by the integrable model. Recall that in quantum models, fermionic and bosonic modes give rise to different free energy functions involved in TBA [40], and related statistical factors appearing, for instance, in Euler-scale correlation functions [18] (although it is sometimes possible to use both formulations to describe the same system [52]). Likewise, it has been observed that classical gases are associated to Boltzmann-type free energy functions [18]. In classical integrable field theory, it is observed in [48–51] that many models admit two types of modes: solitonic modes and "radiative modes", both of which are involved in (generalized) thermalization processes. Solitons have clear particle-like behaviour, giving GHD modes with Boltzmann-like free energy function, like classical gases. By contrast, radiative modes do not display any obvious quasi-particle character in their dynamics, yet, as we will verify, at the level of Euler hydrodynamics, they can be accounted for by the same quasi-particle GHD equations, with the appropriate free energy function for radiative modes. In this paper we only study radiative modes, as the classical sinh-Gordon model does not contain soliton modes.

The radiative free energy function is reminiscent of that of the classical prediction for black-body radiation. Because of the "UV catastrophe", radiative modes make averages of fields containing high enough derivatives diverge. This is characteristic of the roughness of (generalized-)thermally fluctuating classical fields. We will verify that the GHD equations nevertheless describe correctly the averages of observables that are finite. In thermal states of the sinh-Gordon model, the expectation value and correlation functions of every individual

conserved density or current is UV divergent. For our studies of the partitioning protocol and of correlation functions in thermal states, we will therefore focus on the trace of the stress-energy tensor, a combination of energy density and pressure that is UV finite, and more generally on vertex operators (suitable exponentials of the field). In GGEs containing the first (spin-3) non-trivial conserved charge, the energy density and current are UV finite, and thus we also study the partitioning protocol between two such GGEs and evaluate exact profiles of energy density and current.

Our study of vertex operators is based on explicit GGE expectation values obtained via the semi-classical limit of an ansatz recently proposed in the quantum case [53–55]. Therefore, the present work also provides the first numerical test of this ansatz, albeit within the classical framework. Since vertex operators are generically not conserved densities or currents, their correlation functions go slightly beyond the original proposition made in [18]. However, the hydrodynamic projection methods used in [18] can be applied to non-conserved fields (see [25]), and the present work provides the first numerical verification of such concepts.

The paper is organized as follows: Section 2 presents a summary of the hydrodynamic concepts applied to integrable systems, with particular emphasis on the similarities and differences among quantum and classical systems. Section 3 presents numerical results in the classical Sinh Gordon model and comparisons with analytical predictions from GHD. In particular, Section 3.2 studies the partitioning protocol, and Section 3.3 considers correlation functions at the Euler scale in homogeneous thermal ensembles. Our conclusions are gathered in Section 4. Finally, three appendices contain more technical details. Appendix A considers the classical sinh-Gordon model as the semi-classical limit of its quantum counterpart, while appendix B mainly supports Section 3.3 providing the correlation functions on the Eulerian scale in the non-interacting limit. Finally, appendix C contains the numerical methods needed to solve the GHD and to directly simulate the model.

## 2 GHD of classical fields

GHD, and hydrodynamics in general, is based on the assumption of local entropy maximization. It is therefore naturally associated with statistical distributions, such as Gibbs and generalized Gibbs ensembles. Paralleling works done in the quantum case, the natural context in which GHD could be applied to classical fields is not that of the evolution of a single field configuration, but instead that of the deterministic evolution of statistical distributions thereof. This has been considered for instance for anharmonic chains recently [56], where Gibbs ensembles give probability distributions for initial chain configurations, and conventional hydrodynamics successfully describes the evolution of local averages under deterministic dynamics.

Although most of the studies of classical integrable field theory concentrate on exact solutions of the field equations from single field configuration, such as solitonic solutions, there has been important work on statistical distributions of integrable fields as well. There are two categories of results: *i)* those dealing with distributions of solitons [44–47], and *ii)* those dealing with Gibbs ensembles of classical fields [48–51]. Soliton gases, at the Euler scale, also satisfy hydrodynamic equations, which were in fact observed to be those of GHD in [17]. Hence one may expect the GHD of soliton gases to play a role in the hydrodynamics of classical field theory with solitonic modes. However, Gibbs ensembles of classical fields display, in general, both solitonic and radiative modes [48–51]. Below we extend these to generalized Gibbs ensembles (GGEs) (this is a very simple extension). From this we then obtain, using the same hydrodynamic principles as those recalled in [7,8], GHD for classical fields. The solitonic part agrees with the hydrodynamics of soliton gases, but the radiative part gives important extra contributions.

## 2.1 GGEs

We consider a field theory (relativistic or Galilean) with Hamiltonian $H[\Phi, \Pi]$, depending on canonically conjugate fields $\Phi(x)$ and $\Pi(x)$. For the purpose of the general discussion, these may take values in $\mathbb{R}$, in $\mathbb{C}$ or in any other target space.

### 2.1.1 Formulation

This theory is assumed to be integrable, with a space of conserved charges in which $H[\Phi, \Pi]$ lies, spanned by some basis $Q_i[\Phi, \Pi]$ for $i \in \mathbb{N}$. In the conventional view on integrability, this would be taken as a basis of local conserved charges, which can be written as

$$Q_i[\Phi, \Pi] = \int \mathrm{d}x \, \mathfrak{q}_i(x),$$

where $\mathfrak{q}_i(x)$ is a local functional of $\Phi$ and $\Pi$ at $x$ (that is, involving these fields or their derivatives and products thereof at the point $x$). The modern viewpoint also admits the so-called quasi-local conserved charges. These have densities $\mathfrak{q}_i(x)$ with "finite support" (usually with exponentially decaying tails away from $x$). They have been fully understood in the quantum XXZ spin chain [57–59], and, completing with respect to an appropriate inner product on the space of conserved charges, they generate the Hilbert space of pseudo-local charges, rigorously shown in quantum lattices to be involved in generalized thermalization [60]. The general picture is similar in the case of quantum field theory, although the understanding of non-local conserved charges has not reached the same level of maturity [61–68]. It is natural to expect that quasi-local charges should also be involved in GGEs arising from non-equilibrium classical field theory. For the present purpose, we simply follow the standard arguments of GHD [7,8] and assume appropriate completeness of $\{Q_i[\Phi, \Pi]\}$ in order to obtain GHD equations.

We are interested in generalized Gibbs ensembles. These are statistical distributions of field configurations with averages $\langle \cdots \rangle$ described formally as[1]

$$\langle \mathcal{O}_1(x_1) \cdots \mathcal{O}_N(x_N) \rangle = \frac{\int \mathscr{D}\Phi \mathscr{D}\Pi \, \mathcal{O}_1(x_1) \cdots \mathcal{O}_N(x_N) \, e^{-\sum_i \beta_i Q_i[\Phi, \Pi]}}{\int \mathscr{D}\Phi \mathscr{D}\Pi \, e^{-\sum_i \beta_i Q_i[\Phi, \Pi]}} \,, \tag{1}$$

for $\mathcal{O}_k(x_k)$ any (quasi-)local functional of $\Phi$ and $\Pi$ at $x_k$. Gibbs ensembles at inverse temperature $\beta$, with $\beta H[\Phi, \Pi]$ instead of $\sum_i \beta_i Q_i[\Phi, \Pi]$ in (1), were studied in the sine-Gordon model [48–50] and sinh-Gordon model [51] using the classical inverse scattering method, and in some generality using the semi-classical limit of the quantum Thermodynamic Bethe ansatz [69]. These studies gave rise to expressions for the free energy that closely resemble those found in the thermodynamic Bethe ansatz of quantum integrable models [39, 40]. Extending to generalized Gibbs ensembles, this in turns allows for the evaluation of averages $\langle \mathfrak{q}_i \rangle$ of (quasi-)local conserved densities in classical GGEs (here and below, a field written without explicit space-time position is assumed to be at the origin).

Under the evolution determined by the equations of motion of the field theory in the usual fashion, conserved densities satisfy the conservation equation

$$\partial_t \mathfrak{q}_i(x, t) + \partial_x \mathfrak{j}_i(x, t) = 0. \tag{2}$$

The currents $\mathfrak{j}_i(x)$ are also (quasi-)local functionals of $\Phi$ and $\Pi$ at $x$. Their averages in GGEs play a fundamental role in the hydrodynamic description. These do not immediately follow

---

[1]We remark that, although this ensemble is intuitively clear, it is formal as one would need to specify a convergence condition for $\sum_i \beta_i Q_i[\Phi, \Pi]$, in addition to an appropriate definition of the measure of integration over field configurations. Paralleling the quantum context, the correct prescription is expected to be that the conserved charge $W[\Phi, \Pi] = \sum_i \beta_i Q_i[\Phi, \Pi]$ be pseudo-local. Again, the precise meaning of this statement will not play any role below.

from the free energy. However, the point of view in which the classical field theory is the semi-classical limit of a quantum field theory – which is certainly valid in the sinh-Gordon and sine-Gordon models for instance – allows us to take the fundamental results of [7,8] in order to arrive directly at expressions for the averages $\langle j_i \rangle$ of conserved currents in classical field theory.

### 2.1.2 The quasi-particle description

We now state the main results. For the purpose of clarifying the structure, we first put the results within a general framework, abstracting the structure of the formulae found in [48–51] for the sine-Gordon and sinh-Gordon models. As per the above comments, these are natural extensions to GGEs, and to averages of currents, of the formulae found there.

In the present viewpoint, the inverse-scattering solutions of the field theory involves a set $\mathscr{A}$ of possible quasi-particles. Here we use the "quasi-particle" terminology in a loose way: each quasi-particle may be of solitonic or radiative type, the former representing isolated poles in the scattering data, the latter singularities that are part of a continuum (such as a branch cut). Quasi-particles are also characterized by rapidities lying in $\mathbb{R}$, which in the Galilean case can be thought of instead as velocities. The doublet $\boldsymbol{\theta} = (\theta, a) \in \mathbb{R} \times \mathscr{A} = \mathscr{S}$ can be seen as a "spectral parameter" for the quasi-particle, and $\mathscr{S}$ the spectral space. An inverse scattering solution is fully determined by giving a set $\{\boldsymbol{\theta}_\ell : \ell = 1, \dots, L\}$ of spectral parameters of solitonic type, and densities $\rho(\boldsymbol{\theta})$ over spectral parameters of radiative type. This has the physical meaning that the solution contains these quasi-particles (solitons and radiative modes) with these rapidities and densities. Each conserved charge $Q_i[\Phi, \Pi]$ is fully characterized by a spectral function $h_i(\boldsymbol{\theta})$. This is in the sense that, when evaluated in a field configuration $[\Phi, \Pi]$ corresponding to the set $\{\boldsymbol{\theta}_\ell\}$ and the densities $\rho(\boldsymbol{\theta})$, it takes the value

$$Q_i[\Phi, \Pi] = \sum_\ell h_i(\boldsymbol{\theta}_\ell) + \sum_a \int \mathrm{d}\theta \, \rho(\theta, a) h_i(\theta, a), \tag{3}$$

where the sum over $a$ is only over quasi-particle type $a$ of radiative type, and the spectral parameters $\boldsymbol{\theta}_\ell$ are all of solitonic type. For instance, a model composed of $N$ independent free fields contains $N$ radiative modes and no solitonic modes, and the densities $\rho(\theta, a)$ (for $a = 1, 2, \dots, N$) are simply related to the Fourier transforms of the fields. By contrast, the sine-Gordon model contains two solitonic modes (the soliton and anti-soliton) and one radiative mode. See [69–73] for details of the inverse scattering method.

### 2.1.3 Averages of densities and currents

Recall that a GGE is characterized by a set of generalized inverse temperatures $\{\beta_i\}$. It was found in [48–51] that, as in the quantum case, the main object for the thermodynamics of classical field theory is the pseudo-energy $\epsilon(\boldsymbol{\theta})$, which depends on a spectral parameter $\boldsymbol{\theta}$. This solves the following integral equation [48–51]:

$$\epsilon(\theta, a) = w(\theta, a) - \sum_{b \in \mathscr{A}} \int_{\mathbb{R}} \frac{\mathrm{d}\gamma}{2\pi} \varphi_{a,b}(\theta, \gamma) F_a(\epsilon(\gamma, b)), \tag{4}$$

where the source term is determined by the generalized inverse temperatures as

$$w(\boldsymbol{\theta}) = \sum_i \beta_i h_i(\boldsymbol{\theta}), \tag{5}$$

and where $F_a(\epsilon)$ is the free energy function, which is a characteristic of the type of mode that the quasi-particle $a$ represents:

$$F_a(\epsilon) = \begin{cases} e^{-\epsilon} & (a \text{ is a solitonic mode}) \\ -\log \epsilon & (a \text{ is a radiative mode}). \end{cases} \tag{6}$$

For the sake of comparison, recall that the pseudo-energy of the quantum TBA is of a similar form, where the free energy function is instead

$$F_a(\epsilon) = \begin{cases} \log(1 + e^{-\epsilon}) & (a \text{ is a quantum fermion}) \\ -\log(1 - e^{-\epsilon}) & (a \text{ is a quantum boson}). \end{cases} \tag{7}$$

In (4), the quantity $\varphi_{a,b}(\theta, \gamma)$, which we will also denote by $\varphi(\boldsymbol{\theta}, \boldsymbol{\gamma})$ for $\boldsymbol{\theta} = (\theta, a)$ and $\boldsymbol{\gamma} = (\gamma, b)$, is the "differential scattering phase", which characterizes the interactions between the quasi-particles (we assume it to be symmetric). See [48–51] for its explicit form in the sine-Gordon and sinh-Gordon models (the latter is recalled in the next section).

As usual, the model is further specified by two spectral functions: the momentum $p(\boldsymbol{\theta})$ and the energy $E(\boldsymbol{\theta})$ (e.g. in relativistic models $p(\theta, a) = m_a \sinh\theta$ and $E(\theta, a) = m_a \cosh\theta$ where $m_a$ is the mass of the quasi-particle $a$). Averages of local conserved densities are evaluated by differentiating the free energy

$$\mathscr{F} = \sum_{a \in \mathscr{A}} \int_{\mathbb{R}} \frac{d\theta}{2\pi} p'(\theta, a) F_a(\epsilon(\theta, a)) \tag{8}$$

with respect to $\beta_i$ (here and below the prime means rapidity derivative, e.g. $p'(\theta, a) = \partial p(\theta, a)/\partial\theta$). Averages of currents are similarly obtained by differentiating [7]

$$\mathscr{G} = \sum_{a \in \mathscr{A}} \int_{\mathbb{R}} \frac{d\theta}{2\pi} E'(\theta, a) F_a(\epsilon(\theta, a)). \tag{9}$$

The results are

$$\langle \mathsf{q}_i \rangle = \int_{\mathscr{S}} \frac{d\boldsymbol{\theta}}{2\pi} p'(\boldsymbol{\theta}) n(\boldsymbol{\theta}) h_i^{\mathrm{dr}}(\boldsymbol{\theta}) = \int_{\mathscr{S}} d\boldsymbol{\theta}\, \rho_{\mathrm{p}}(\boldsymbol{\theta}) h_i(\boldsymbol{\theta}) \tag{10}$$

and

$$\langle \mathsf{j}_i \rangle = \int_{\mathscr{S}} \frac{d\boldsymbol{\theta}}{2\pi} E'(\boldsymbol{\theta}) n(\boldsymbol{\theta}) h_i^{\mathrm{dr}}(\boldsymbol{\theta}) = \int_{\mathscr{S}} d\boldsymbol{\theta}\, v^{\mathrm{eff}}(\boldsymbol{\theta}) \rho_{\mathrm{p}}(\boldsymbol{\theta}) h_i(\boldsymbol{\theta}), \tag{11}$$

where we use the notation

$$\int_{\mathscr{S}} \frac{d\boldsymbol{\theta}}{2\pi} = \sum_{a \in \mathscr{A}} \int_{\mathbb{R}} \frac{d\theta}{2\pi} \qquad (\text{for } \boldsymbol{\theta} = (\theta, a)). \tag{12}$$

In these expressions, we have introduced the usual TBA quantities and operations: the occupation function

$$n(\theta, a) = -\left.\frac{\partial F_a(\epsilon)}{\partial\epsilon}\right|_{\epsilon = \epsilon(\theta, a)} = \begin{cases} e^{-\epsilon(\theta, a)} & (a \text{ is a solitonic mode}) \\ \dfrac{1}{\epsilon(\theta, a)} & (a \text{ is a radiative mode}), \end{cases} \tag{13}$$

the dressing operation

$$h^{\mathrm{dr}}(\boldsymbol{\theta}) = h(\boldsymbol{\theta}) + \int_{\mathscr{S}} \frac{d\boldsymbol{\gamma}}{2\pi} \varphi(\boldsymbol{\theta}, \boldsymbol{\gamma}) n(\boldsymbol{\gamma}) h^{\mathrm{dr}}(\boldsymbol{\gamma}), \tag{14}$$

the quasi-particle density

$$2\pi\rho_{\mathrm{p}}(\boldsymbol{\theta}) = (p')^{\mathrm{dr}}(\boldsymbol{\theta})\, n(\boldsymbol{\theta}), \tag{15}$$

and the effective velocity

$$v^{\mathrm{eff}}(\boldsymbol{\theta}) = \frac{(E')^{\mathrm{dr}}(\boldsymbol{\theta})}{(p')^{\mathrm{dr}}(\boldsymbol{\theta})}. \tag{16}$$

Again, (13) is to be contrasted with the expressions in the quantum case,

$$n(\theta, a) = \begin{cases} \dfrac{1}{e^{\epsilon(\theta,a)}+1} & (a \text{ is a quantum fermion}) \\ \dfrac{1}{e^{\epsilon(\theta,a)}-1} & (a \text{ is a quantum boson}). \end{cases} \tag{17}$$

The second expression on the right-hand side in (10) is to be compared with the expression (3) of the conserved charge on a single field configuration: we see that the solitonic modes are now also described using densities, representing the fact that we have a thermodynamically large number of solitons.

## 2.2 GHD

Given that the structure of GGEs in classical field theory is the same as that for quantum models, up to a modification of the free energy function, it follows that all GHD results can immediately be adapted to the hydrodynamic of classical fields, under the same hydrodynamic assumption. That is, consider local observables in states with weak, large-scale inhomogeneities:

$$\langle \mathcal{O}(x,t)\rangle_{\mathrm{inhomogeneous}} = \frac{\int \mathscr{D}\Phi \mathscr{D}\Pi\, \mathcal{O}(x,t)\, e^{-\int \mathrm{d}y \sum_i \beta_i(y)\mathsf{q}_i(y)}}{\int \mathscr{D}\Phi \mathscr{D}\Pi\, e^{-\int \mathrm{d}y \sum_i \beta_i(y)\mathsf{q}_i(y)}}. \tag{18}$$

Here the time-evolved field $\mathcal{O}(x,t)$ is obtained from the equations of motion of the field theory in the usual fashion. If $\beta_i(y)$ vary non-trivially only on very large scales, the main assumption is that

$$\langle \mathcal{O}(x,t)\rangle_{\mathrm{inhomogeneous}} = \langle \mathcal{O}\rangle_{x,t}, \tag{19}$$

with space-time dependent GGEs,

$$\langle \cdots \rangle_{x,t} = \frac{\int \mathscr{D}\Phi \mathscr{D}\Pi\, (\cdots)\, e^{-\sum_i \beta_i(x,t)Q_i[\Phi,\Pi]}}{\int \mathscr{D}\Phi \mathscr{D}\Pi\, e^{-\sum_i \beta_i(x,t)Q_i[\Phi,\Pi]}}. \tag{20}$$

The large-scale conservation laws $\partial_t \langle \mathsf{q}_i\rangle_{x,t} + \partial_x \langle \mathsf{j}_i\rangle_{x,t} = 0$ give the GHD equations. In terms of the fluid variable $n(x,t;\boldsymbol{\theta})$ this boils down to [7,8]

$$\partial_t n(x,t;\boldsymbol{\theta}) + v^{\mathrm{eff}}(x,t;\boldsymbol{\theta})\partial_x n(x,t;\boldsymbol{\theta}) = 0. \tag{21}$$

Combined with the description of GGEs in the previous subsection, this is the Euler hydrodynamic equation for classical field theory.

We can now immediately apply the various results from GHD. Below, we will make use of two sets of results: those pertaining to the partitioning protocol [7,8], and those relating to Euler-scale correlation functions in GGEs [18, 20, 25]. The former give averages in non-equilibrium states where non-zero currents exist, while for the latter we will restrict ourselves to dynamical correlations in homogeneous, stationary states (see [25] for the case of inhomogeneous, non-stationary states). We briefly review the main results.

### 2.2.1 Partitioning protocol

In the partitioning protocol, the initial state has probability measure

$$\exp\left[-\int_{y<0} dy \sum_i \beta_i^L \mathfrak{q}_i(y) - \int_{y>0} dy \sum_i \beta_i^R \mathfrak{q}_i(y)\right]. \tag{22}$$

This is not of weak variation, as there is a sharp jump at the origin. However, after a small relaxation time, it has been observed both in quantum spin chains [8] and in classical gases [13] that the generalized hydrodynamic description applies, with the corresponding hydrodynamic initial state. We thus set $n(x,0;\boldsymbol{\theta}) = n_L(\boldsymbol{\theta})\Theta(-x) + n_R(\boldsymbol{\theta})\Theta(x)$ (where $\Theta(x)$ is Heavyside's step function), which is composed of a GGE on the left-hand side, $n_L(\boldsymbol{\theta})$, and a GGE on the right-hand side, $n_R(\boldsymbol{\theta})$, and we solve (21) with this initial condition. The solution depends only on the ray $\zeta = x/t$. It is obtained by solving the following "self-consistent" set of integral equations, for the functions $n(\zeta;\boldsymbol{\theta})$ and $v^{\text{eff}}(\zeta,\boldsymbol{\theta})$:

$$n(\zeta;\boldsymbol{\theta}) = n_L(\boldsymbol{\theta})\Theta(v^{\text{eff}}(\zeta,\boldsymbol{\theta}) - \zeta) + n_R(\boldsymbol{\theta})\Theta(\zeta - v^{\text{eff}}(\zeta,\boldsymbol{\theta})). \tag{23}$$

Here, $v^{\text{eff}}(\zeta;\boldsymbol{\theta})$ is determined by (16) where the dressing is with respect to the occupation function $n(\zeta;\boldsymbol{\theta})$. This set of equations was found to be solvable rather efficiently by iteration [7,8].

### 2.2.2 Euler-scale correlations

GHD also provides exact Euler-scale correlation functions, which give information about connected correlation functions

$$\langle \mathcal{O}_1(x,t)\mathcal{O}_2(0,0)\rangle^c = \langle \mathcal{O}_1(x,t)\mathcal{O}_2(0,0)\rangle - \langle \mathcal{O}_1(x,t)\rangle\langle \mathcal{O}_2(0,0)\rangle$$

in homogeneous, stationary GGEs at large scales [18,20,25]. We find that the precise definition of Euler-scale correlation functions involve appropriate averaging over fluid cells (see also [25]). We numerically observe below and explicitly test in the free field limit (see Appendix B) that such an averaging is indeed necessary in order to obtain results predicted by the theory of Euler hydrodynamics. We expect there to be many ways of doing such averaging. Consider for instance a neighbourhood $\mathcal{N}_\lambda(x,t)$ around the point $(\lambda x, \lambda t)$, which can be thought of as a disk of radius $r_\lambda$ that grows with $\lambda$ fast enough, but under the condition $\lim_{\lambda\to\infty} r_\lambda/\lambda = 0$. Denote its area by $|\mathcal{N}_\lambda|$. Then, GHD predicts

$$\lim_{\lambda\to\infty} \lambda \int_{\mathcal{N}_\lambda(x,t)} \frac{dy\, d\tau}{|\mathcal{N}_\lambda|} \langle \mathcal{O}_1(y,\tau)\mathcal{O}_2(0,0)\rangle^c = \int d\boldsymbol{\theta}\, \delta(x - v^{\text{eff}}(\boldsymbol{\theta})t)\rho_p(\boldsymbol{\theta})f(\boldsymbol{\theta})V^{\mathcal{O}_1}(\boldsymbol{\theta})V^{\mathcal{O}_2}(\boldsymbol{\theta}). \tag{24}$$

Here $f(\boldsymbol{\theta})$ is the statistical factor, which equals

$$f(\theta,a) = -\frac{\partial^2 F_a(\epsilon)/\partial\epsilon^2}{\partial F_a(\epsilon)/\partial\epsilon}\bigg|_{\epsilon=\epsilon(\theta,a)} = \begin{cases} 1 & (a \text{ is a solitonic mode}) \\ n(\theta,a) & (a \text{ is a radiative mode}). \end{cases} \tag{25}$$

This is again to be compared with the quantum case,

$$f(\theta,a) = \begin{cases} 1 - n(\theta,a) & (a \text{ is a quantum fermion}) \\ 1 + n(\theta,a) & (a \text{ is a quantum boson}). \end{cases} \tag{26}$$

The functions $V^{\mathcal{O}}$ are obtained from the hydrodynamic projection

$$\langle Q_j \mathcal{O}\rangle = -\frac{\partial}{\partial\beta_i}\langle \mathcal{O}\rangle = \int_{\mathcal{S}} d\boldsymbol{\theta}\, h_j^{\text{dr}}(\boldsymbol{\theta})\rho_p(\boldsymbol{\theta})f(\boldsymbol{\theta})V^{\mathcal{O}}(\boldsymbol{\theta}), \tag{27}$$

which is valid for any local observable. In the case of charge densities and currents, the explicit expressions are $V^{q_i}(\boldsymbol{\theta}) = h_i^{dr}(\boldsymbol{\theta})$ and $V^{j_i}(\boldsymbol{\theta}) = v^{eff}(\boldsymbol{\theta})h_i^{dr}(\boldsymbol{\theta})$.

Of course the integral in (24) can be performed and is supported, thanks to the delta function, on the values of $\boldsymbol{\theta}$ for which $v^{eff}(\boldsymbol{\theta}) = x/t$. Integrating over space $x$, the $\lambda$ factor disappears and we obtain

$$\lim_{t \to \infty} \int_{\mathcal{N}_t(\tau)} \frac{d\tau}{|\mathcal{N}_t|} \int dx \, \langle \mathcal{O}_1(x, \tau)\mathcal{O}_2(0, 0)\rangle^c = \int d\boldsymbol{\theta} \, \rho_p(\boldsymbol{\theta})f(\boldsymbol{\theta})V^{\mathcal{O}_1}(\boldsymbol{\theta})V^{\mathcal{O}_2}(\boldsymbol{\theta}), \qquad (28)$$

where we keep the time-domain fluid-cell average, with $\mathcal{N}_t(\tau)$ an interval of radius $r_t$ centred on $\tau$. See [18, 25] for more details.

**Remark.** As mentioned above, the GHD of classical fields can be seen as composed, in general, of two types of "quasi-particles": solitonic modes and radiative modes. All aspects of GHD that pertain to the solitonic modes – such as the solitonic part of the formula for effective velocity and the statistical factors in correlation functions – agree *exactly* with the corresponding aspects of the GHD for gases of solitons. That is, there is a part of the hydrodynamics which arises from the gas of the field theory's own solitons. However, this is *not enough* to describe correctly the hydrodynamics, as the radiative modes are also necessary. We also emphasize that the statistical factor of solitonic modes agrees with the statistical factor for particle gases, such as the hard-rod gas [18]. Thus solitons truly behave as particles, while radiative modes do not.

# 3 The sinh-Gordon model

The results presented above follow from *i)* generalizing the results of [51, 69], for the sine-Gordon and sinh-Gordon models, to arbitrary models in arbitrary GGEs, and *ii)* combining with the results of [7, 8] in order to get the hydrodynamics. The generalization to GGEs is a straightforward exercise, and the abstraction to arbitrary models is a natural guess. The expressions for the current averages $\langle j_i \rangle$ in (11), and the effective velocity (16), were never written down before in classical field theory. However, they are again natural from the fact that GGEs, in the quasi-particle formulation, have a very similar structure in classical and quantum field theories. For completeness we present in Appendix A a full derivation of the hydrodynamics via the semi-classical limit of the quantum sinh-Gordon model, following the ideas of [69]. Since in the quantum sinh-Gordon model the corresponding equations, and in particular the effective velocity (16), were derived in [7], this gives a full derivation of all necessary ingredients for the hydrodynamics of the classical sinh-Gordon model (it would be interesting to have an independent derivation of the effective velocity fully within classical field theory, which we leave for later works).

An advantage of dealing with a classical theory, as compared with a quantum model, is the relative simplicity of testing it through numerical simulations. This is especially true for continuous models, where the DMRG methods [74] are not efficient. In this section, after specializing the hydrodynamics to the classical sinh-Gordon model, we test its predictions against direct numerical simulations, whose details are left to Appendix C.

### 3.1  GGE of the sinh-Gordon model and UV finiteness

The sinh-Gordon model (ShG) is described in terms of a single scalar field $\Phi$ whose dynamics is ruled by the following Lagrangian:

$$\mathcal{L}_{\mathrm{ShG}} = \frac{1}{2}\partial_\mu \Phi \partial^\mu \Phi - \frac{m^2}{g^2}(\cosh(g\Phi) - 1). \tag{29}$$

We will be interested in particular in the canonical stress energy tensor $T_{\mu\nu}$,

$$T_{\mu\nu} = -\eta_{\mu\nu}\mathcal{L} + \partial_\mu \Phi \frac{\delta\mathcal{L}}{\delta\partial^\nu\Phi}, \qquad T^\mu_{\ \mu} = 2\frac{m^2}{g^2}(\cosh(g\Phi) - 1) \tag{30}$$

with $\eta_{\mu\nu}$ the Minkowski two-dimensional metric.

#### 3.1.1  GGEs

The model is shown to be classically integrable by means of the inverse scattering method [69–73] and it is a well known fact that its integrability survives under quantization [75, 76]. At the classical level the spectrum of ShG possesses only one purely radiative species [69], making its TBA and the subsequent hydrodynamics rather simple. The resulting pseudo-energy equation [51,69] can be brought to the form (4), in the case of radiative mode as per (6), with $\varphi(\theta, \gamma) = (g^2/4)\cosh(\theta - \gamma)/\sinh^2(\theta - \gamma)$. Equivalently, it is

$$\epsilon(\theta) = w(\theta) - \int \frac{\mathrm{d}\gamma}{2\pi}\tilde{\varphi}(\theta, \gamma)\partial_\gamma \log\epsilon(\gamma), \tag{31}$$

with

$$\tilde{\varphi}(\theta, \gamma) = \frac{g^2}{4}\frac{1}{\sinh(\theta - \gamma)}. \tag{32}$$

Recall that the source term $w(\theta)$ determines the particular GGE state considered. The integral in (31) is a Cauchy principal value integral for the singularity at $\theta = \gamma$. In [69], (31), (32) are derived both through the inverse scattering method and the semi-classical limit of the quantum TBA[2]. For completeness, in Appendix A we report the semi-classical computation, together with the derivation of the full hydrodynamics.

In general, local conserved charges can be organised into irreducible representations of the 1+1-dimensional Lorentz group, and, as is often done, these can be combined into parity-odd and parity-even charges. In the sinh-Gordon model, all odd spins arise, but no even spins. In the following we concentrate on the parity-even charges, which we denote $H_n$ for $n \in \mathbb{N}$ odd. These have spectral functions $h_n(\theta)$ given by

$$h_n(\theta) = m^n \cosh(n\theta).$$

In particular, the Hamiltonian is $H_1 = H = \int \mathrm{d}x\, T^{00}(x)$. Thus a GGE containing only local parity-even conserved quantities is described by the path integral (1) with weight

$$\exp\Big[-\sum_{n\in\mathbb{N}\ \mathrm{odd}} \beta_n H_n\Big], \tag{33}$$

and determined within TBA by the source term

$$w(\theta) = \sum_{n\in\mathbb{N}\ \mathrm{odd}} \beta_n h_n(\theta). \tag{34}$$

---

[2] A typo of an overall minus sign in the kernel occurred in [69].

Although quasi-local charges are not fully worked out in the sinh-Gordon model, we expect they largely extend the space of allowed functions $w(\theta)$. Generically, the states emerging from the GHD time evolution belong to such an extended space.

Note that the Lagrangian density is invariant under simultaneous scaling

$$(x, t, m, \Phi, g) \mapsto (\lambda x, \lambda t, \lambda^{-1} m, \lambda \Phi, \lambda^{-1} g),$$

while it gains a factor $\lambda^{-2}$ under $(x, t, m) \mapsto (\lambda x, \lambda t, \lambda^{-1} m)$. Hence, from the latter, the GGE state (33) is invariant under

$$(x, t, m, g, \{\beta_n\}) \mapsto (\lambda x, \lambda t, \lambda^{-1} m, g, \{\lambda^n \beta_n\}), \tag{35}$$

while from the former (and the explicit form of higher conserved densities, see Appendix C.2), it is invariant under

$$(x, t, m, g, \{\beta_n\}) \mapsto (\lambda x, \lambda t, \lambda^{-1} m, \lambda^{-1} g, \{\lambda^{n-2} \beta_n\}). \tag{36}$$

As a consequence, in situations that only depend on the ratio $x/t$ (as in the partitioning protocol and in the study of correlation functions in homogeneous states, considered below), the invariant ratios characterizing the strength of the interaction are

$$\frac{g^2}{\beta_n m^n}, \quad n \in \mathbb{N}.$$

In particular, the effective interaction strength is larger at large temperatures.

### 3.1.2 UV finiteness

It is well known that thermal states in classical electromagnetism suffer from UV catastrophes, intimately connected to Planck's original idea of quantizing oscillation modes in order to explain the black body radiation spectrum. The same problem appears in thermal ensembles and GGEs of other field theories such as the sinh-Gordon model. The problem is that in such states fields can be very "rough". For instance, a GGE (33) involving local conserved charges only up to some finite spin (where the sum is finite) does not regularize the field enough to guarantee the existence of averages of its large-order derivatives. In the TBA formulae above, this is clearly seen as a divergence of averages of conserved densities of high enough spin, which occur at large rapidities due to radiative modes. Indeed, consider (10) and (11) in the sinh-Gordon model. At large rapidities, $|v^{\text{eff}}(\theta)| \sim |\tanh \theta| \sim 1$ is bounded. However, $2\pi \rho_{\text{p}}(\theta) = m \cosh^{\text{dr}}(\theta) n(\theta) \sim (1/2) e^{|\theta|}/\epsilon(\theta) \sim (1/2) e^{|\theta|}/w(\theta)$ where $w(\theta)$ is the source term in (4), while $h^{\text{dr}}(\theta) \sim h(\theta)$. Thus, convergence of the average of the density or current corresponding to $h(\theta)$ is guaranteed only if $h(\theta) e^{|\theta|}/w(\theta)$ is integrable. In a thermal ensemble, $w(\theta) \sim (\beta/2) e^{|\theta|}$ (where $\beta$ is the inverse temperature), and therefore no local conserved density or current has finite average (except for those that are zero by parity symmetry). The same conclusion holds in the nontrivial ensemble emerging from the partitioning protocol from two thermal baths, as it has the same asymptotics in $\theta$ as those of the original baths. Nevertheless, it is a simple matter to see from the TBA formulae that the right combination of energy density and momentum current is finite: this is the trace of the stress-energy tensor $T^\mu_\mu$, which does not contain field derivatives. Clearly, averages of higher-spin densities and currents become finite as we increase the spin of the conserved charges involved in the GGE.

Vertex operators $e^{kg\Phi}$ also have finite thermal averages, as they do not contain field derivatives. Since GHD provides the full GGE state at every space-time point – not just the averages of conserved densities and currents – we can combine GHD information together with homogeneous GGE results to study these quantities. For instance, following [69], we could access

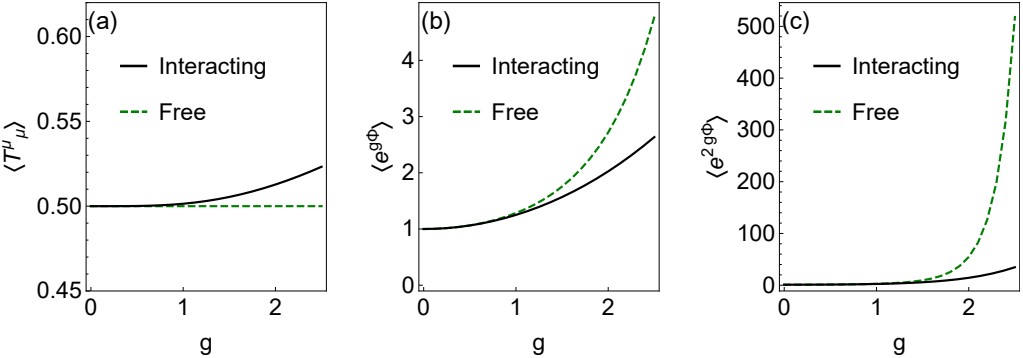

Figure 1: Analytical expectation values of the trace of the stress energy tensor and of vertex operators in a homogeneous thermal ensemble at inverse temperature $\beta$ are compared with the free model. By scaling (35), (36), we may choose $\beta = m = 1$ and vary $g$. First panel on the left: the traces of the stress energy tensor for the interacting and free cases are compared. Second and third panel: the same is done for vertex operators. Note that passing from the interacting to the free theory, the trace of the stress energy tensor changes as function of the field. The difference between interacting and free is enhanced when comparing expectation values of fixed functions of the fields.

their expectation values trough the LeClair-Mussardo expansion [77], but this provides a small excitation-density description, which is not best suited for the present purpose. However, an ansatz for the ratios of expectation values of vertex operators in the *quantum* model, valid for arbitrary excitation density, exists. It has been firstly proposed for thermal ensembles [53,54] and recently extended to arbitrary GGEs [55]. Taking the classical limit, we obtain formulae for expectation values in the classical sinh-Gordon model; this is a new result within the classical realm. The derivation is reported in Appendix A.

## 3.2 The partitioning protocol

In this subsection, we study the classical sinh-Gordon model in the partitioning protocol. We recall that this is the protocol where the initial state is formed of two homogeneous field distributions, one for the left half $x < 0$ and the other for the right half $x > 0$ of space, as per (22). This setup is useful to study as it generates truly non-equilibrium states, with nonzero currents and nontrivial profiles in space-time, yet it is simple enough so that GHD provides easily workable expressions. It is also the setup which is expected to be most accurately described by the hydrodynamic approximation, as at long times, profiles smooth out, making Euler hydrodynamics more applicable (in particular, no shocks are expected to develop, see [7,8]). Thus this setup gives the best playground for verifications of GHD non-equilibrium predictions.

### 3.2.1 Description of the protocol and observables

For simplicity, we consider ensembles with nonzero (generalized) temperatures $\beta_i$ for the Hamiltonian $H = H_1$ and for the first nontrivial (spin-3) parity-even local conserved charge $H_3$ only.

In the partitioning protocol, one needs to specify the boundary or continuity conditions of the fields at $x = 0$. An exact separation of the two halves is possible by considering an independent copy of the model on each half-line, with boundary conditions (such as Dirichlet or von Neumann) at $x = 0^+$ and $x = 0^-$. This is natural in a continuous field theory. However, in order to simulate the partitioning protocol, we of course construct a discretisation of the

sinh-Gordon model. In this case, one may instead explicitly keep the connection between the two halves as follows: we simply sum the densities of Hamiltonian and of other charges over all sites, with coefficients assigned according to Eq. (22), where a change from $\beta^L$ to $\beta^R$ occurs, say, in passing from the site at $x = 0$ to the site immediately to its right. This is better behaved numerically, and this is the choice we have made (see Appendix C). Other regularisations are possible, such as interpolating functions that approach the correct thermal ensembles at large distances. In all cases, the same behaviours at large times are expected, as described by GHD.

Recall that the large-time state is determined, according to GHD, as follows. We use (31) with $n(\theta) = 1/\epsilon(\theta)$ in order to fix the left and right occupation functions $n_L(\theta)$ and $n_R(\theta)$. These are obtained by injecting the source terms $w^{L,R}(\theta)$ as per (34) with the values of $\beta_n = \beta_n^{L,R}$ for $n = 1, 3$ that appear in (22), all other parameters being zero ($\beta_n = 0$ for all $n \neq 1, 3$). Recall that $h_1(\theta) = m \cosh \theta$ and $h_3(\theta) = m^3 \cosh 3\theta$ are the spectral functions of the energy and of the spin-3 charge, respectively. Once $n_L(\theta)$ and $n_R(\theta)$ are fixed, then the large-time state $n(\zeta; \theta)$ is obtained by solving (23).

We study two types of observables. The first type are energy-momentum tensor components. The energy density and current, $T^{00}(x)$ and $T^{10}(x)$ respectively, have averages predicted by (10-11). The trace of the stress-energy tensor $T^\mu_\mu$ – the difference between the energy density and the momentum current – is given by

$$\langle T^\mu_\mu \rangle = m \int \frac{d\theta}{2\pi} \rho_p(\theta) \big( \cosh \theta - v^{\text{eff}}(\theta) \sinh \theta \big) . \tag{37}$$

The second type are generalizations of the trace of the energy-momentum tensory, the vertex operators $e^{kg\Phi}$. Based on Ref. [53–55], a recursive relation for the GGE averages of vertex operators $e^{kg\Phi}$ can be obtained (for details see Appendix A)

$$\frac{\langle e^{(k+1)g\Phi} \rangle}{\langle e^{kg\Phi} \rangle} = 1 + (2k+1)\frac{g^2}{4\pi} \int d\theta \, e^\theta n(\theta) p^k(\theta) , \tag{38}$$

where

$$p^k(\theta) = e^{-\theta} + \frac{g^2}{4} \mathscr{P} \int \frac{d\gamma}{2\pi} \frac{1}{\sinh(\theta - \gamma)} \big( 2k - \partial_\gamma \big) (n(\gamma) p^k(\gamma)) . \tag{39}$$

Taking $k$ to be an integer (and using the fact that $\langle e^{kg\Phi} \rangle = 1$ if $k = 0$), we obtain explicit formulae for $\langle e^{kg\Phi} \rangle$.

Finally, we consider three cases of the partitioning protocol:

I. The purely thermal case, where both sides are at different temperatures ($\beta_1^L \neq \beta_1^R$ and $\beta_3^L = \beta_3^R = 0$, Figs. 2 and 3).

II. The partitioning including differently coupled $H_3$ charges but with equal temperatures ($\beta_1^L = \beta_1^R \neq 0$ and $\beta_3^L \neq \beta_3^R = 0$, Fig. 4).

III. The partitioning where both $H$ and $H_3$ are differently coupled ($\beta_1^L \neq \beta_1^R$ and $\beta_3^L \neq \beta_3^R = 0$, Fig. 5), choosing the energy densities to be equal in the left and right reservoirs.

As explained above, in the purely thermal case (case I), because of the UV catastrophe, the expectation values of energy and momentum densities (as well as all the individual local charge densities and currents) are divergent. Hence in this case we do not have access to the non-equilibrium energy current. However, the averages of the trace of the stress-energy tensor $T^\mu_\mu$ and of the vertex operators $e^{kg\Phi}$ are finite, as these do not contain field derivatives. Thus in the thermal case we concentrate on these observables (Fig. 2). In the cases II and III, which

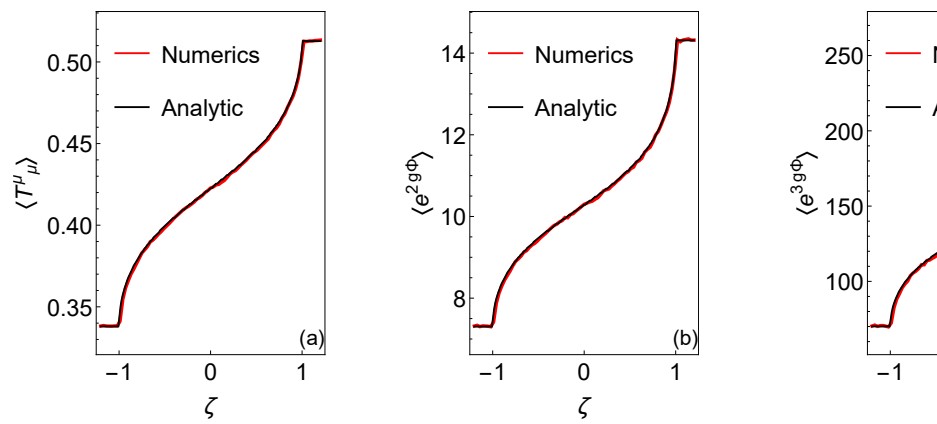

Figure 2: Case I. Profiles of the trace of the stress energy tensor (a) and of the first vertex operators, (b) and (c), as functions of the ray $\zeta = x/t$ for a two-temperature partitioning protocol. Parameters: $m = 1$, $g = 2$, $a = 0.05$ inverse temperatures of the initial ensembles $\beta_1^L = 1.5$ and $\beta_1^R = 1$, the number of samples of the Metropolis is 84000. An additional average along the ray from $t = 50$ up to $t = 75$ smears out the statistical oscillations. In terms of the dimensionless coupling $g^2/m\beta$, we are in the strongly interacting regime (see Figure 1).

include the charge $H_3$, we study the energy density and current independently. Interestingly, we find that a non-equilibrium energy current is generated in case II (Fig. 4). That is, the $H_3$ charge produces an imbalance in energy densities, hence a current develops. Having access to a UV finite non-equilibrium energy current is an important reason for considering the inclusion of $H_3$. In the case III, we study the case where the energy densities are equal on the left and right reservoirs, yet the coupling to $H$ and $H_3$ are not balanced. This emphasizes the effects of the higher conserved charge: an energy current is generated even though the energy densities are balanced in the reservoirs, with a nontrivial profile developing (Fig. 5).

### 3.2.2 Numerical results

In order to illustrate the effect of the interaction and determine the values of $g$ which significantly affect the averages, in Figure 1 we plot the analytical thermal averages of the stress energy tensor and of the vertex operators both in the interacting and the free ($g = 0$) purely thermal Gibbs ensemble at the same temperature $\beta^{-1}$. This indicates that $g^2/(\beta m) \approx 2$ is well into the strongly interacting regime. We then compare the analytic predictions with direct numerical simulations of the above out-of-equilibrium protocol, see Appendix C for details of the numerical methods.

The results of the comparison between GHD analytical predictions and numerical simulations in the thermal case (case I) are presented in Figures 2. We find excellent agreement. Measuring the difference between the numerical and analytical values by the relative $L(1)$ distance

$$R = \frac{\int |v^{\text{num}}(\zeta) - v^{\text{ana}}(\zeta)| d\zeta}{\int |v^{\text{ana}}(\zeta)| d\zeta} \tag{40}$$

(where $v^{\text{num}}$ is the numerical data and $v^{\text{ana}}$ is the analytical value), we obtain $R = 2.1 \times 10^{-3}$, $3.1 \times 10^{-3}$ and $7.1 \times 10^{-3}$ in the cases of Figures 2a, 2b and 2c respectively. It must be stressed that, despite various checks in known limiting cases (see [55]), the formula for the vertex operators (38-39) had never been numerically verified before. Figure 2 constitutes both a verification of the validity of generalized hydrodynamics for the sinh-Gordon model *and* a numerical verification of the classical limit of the vertex operator ansatz.

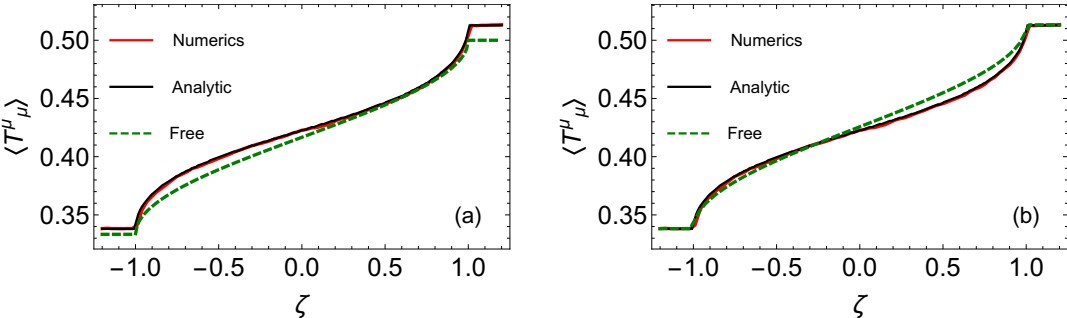

Figure 3: The profile of the trace of the stress energy tensor already reported in Figure 2, both analytic and numerical, with the same parameters used in Figure 2 are compared against the expectation values in the free theory with (a) identical parameters and (b) $\beta_1^L$ and $\beta_1^R$ chosen so that the asymptotic values agree with the interacting theory.

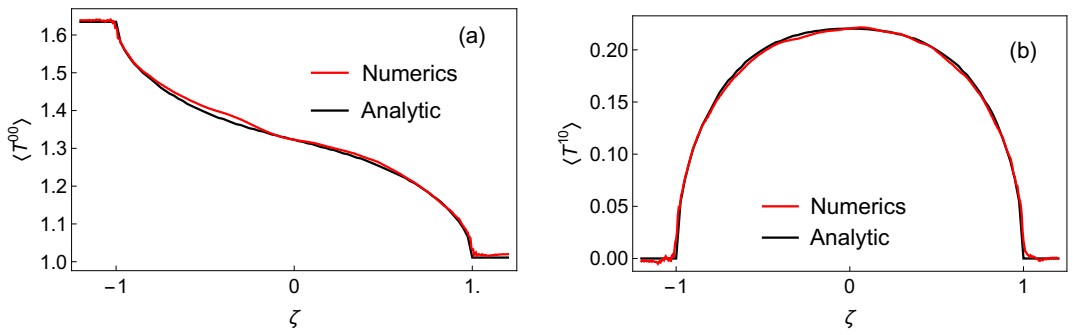

Figure 4: Case II. Profiles of $\langle T^{00}(\zeta)\rangle$ (a) and $\langle T^{10}(\zeta)\rangle$ (b) are numerically computed and compared with the GHD predictions. The numerical results are obtained by averaging over the range $10 \leq t \leq 16$. The parameters are $m = 1$, $g = 1$, $\beta_1^L = \beta_1^R = 1$ with the coupling to the higher-spin charge varying, $\beta_3^L = 1/2$, $\beta_3^R = 1$ and the lattice spacing in the light-cone discretization is $a = 0.1$.

Our data is precise enough to show a clear departure from the free theory result. For comparison, Figure 3 shows the numerical and hydrodynamic curves for the trace of the stress-energy tensor, as well as the curve for the same quantity obtained analytically in the free case ($g = 0$) for two different sets of free parameters. In the first case (a), we use the same values of $\beta_1^L$ and $\beta_2^L$ as in the interacting model, which clearly gives different values of the equilibrium expectation values for $\zeta < -1$ and $\zeta > 1$. We observe a relative $L(1)$ distance $R = 1.5 \times 10^{-2}$ (obtained by replacing $v^{\text{num}}$ by $v^{\text{free}}$ in (40)), thus about one order of magnitude larger compared with distance between numerics and hydrodynamic prediction reported above.

In the second case (b), we take the values of $\beta_1^L$ and $\beta_2^L$ in the free model so that the equilibrium expectation values for $\zeta < -1$ and $\zeta > 1$ agree with those of the interacting model, which from equation (80) means in this case $\beta_a = 1/(2\langle T^\mu{}_\mu\rangle)$, so $\beta_1^L = 1.47842$ and $\beta_1^R = 0.974279$. We still see a clear difference between the free and interacting models and observe a relative $L(1)$ distance $R = 1.3 \times 10^{-2}$, smaller than in case (a) but still an order of magnitude larger compared with distance between numerics and hydrodynamic prediction reported above.

In Figure 4 we make the comparison for the energy density and energy current in the case II, including the charge $H_3$. Again very good agreement is found with the measure of error being $R = 5.0 \times 10^{-3}$ and $1.0 \times 10^{-2}$ for the charge and the current respectively.

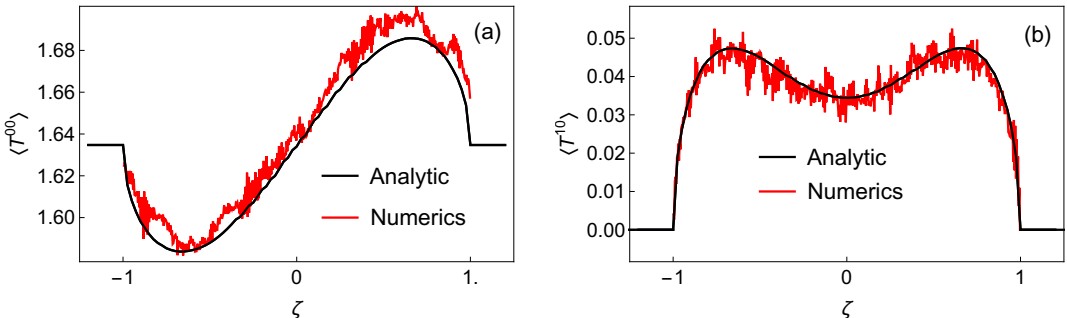

Figure 5: Case III. Profiles of $\langle T^{00}(\zeta)\rangle$ (a) and $\langle T^{10}(\zeta)\rangle$ (b) are numerically computed and compared with the GHD predictions at $t = 25$. The parameters are $m = 1$, $g = 1$, $\beta_1^L = 1$, $\beta_3^L = 1/2$ and $\beta_1^R = 1/2$, $\beta_3^R = 0.609608$ and the lattice spacing in the light-cone discretization is $a = 0.1$.

In Figure 5, we consider an example of case III, with $\beta_1^L \neq \beta_1^R$ and $\beta_3^L \neq \beta_3^R$ but chosen so that the expectation value of the energy density is the same in the two different ensembles. This does not mean that there is no net energy flow - the distribution of energy amongst particles of different speeds is different, and in this example there is a net flow of energy through $x = 0$ at late times. The relative error between the analytic and numerical results is larger in this case because the changes in energy density and the magnitude of the current are an order of magnitude smaller than case II, so the effects of the statistical error and the convergence to the scaling form appear larger on the graph, but they are the same absolute size as those in case II.

Both case II and case III confirm that GHD describes correctly the non-equilibrium energy current emerging from the partitioning protocol.

## 3.3 Euler scale correlation functions

In this subsection, we consider two-point dynamical correlation functions in homogeneous, stationary states (GGEs) of the sinh-Gordon model. For simplicity we consider correlations in homogeneous thermal ensembles, (33) with $\beta_1 \neq 0$ and all other parameters set to 0. Equation (24) gives a prediction for the behaviour of the connected two-point correlators of local observables at the Euler scale: at long times and large distances, after fluid-cell averaging. Such expressions in integrable systems have never been directly compared with numerical simulations and here we present the first study in this direction. We emphasize that Euler-scale correlations are due to all ballistically transported conserved quantities that connect the local observables involved. These generically include all conserved quantities available in the model, this being true also in thermal states and for correlations of the stress-energy tensor components. Thus the present analysis provides good checks of the effects of integrability beyond the conserved energy currents.

Note that there are predictions for Euler correlation functions in inhomogeneous, non-stationary states such as those considered in the previous subsection [25], however these are much more complicated and will be considered in a separate publication.

Again, we take a finite volume on which the system is taken periodic, and analyze correlation functions at times long enough in order to reach the Euler scale, yet small enough in order to avoid finite-volume effects.

We note that the UV catastrophe still prevents us from computing correlation functions of arbitrary charges and currents. However one can check from the TBA formulae that a certain number of stress-energy tensor correlation functions do stay finite, such as $\langle T^{\nu\rho}(x,t)T_\mu^\mu(0,0)\rangle^c$

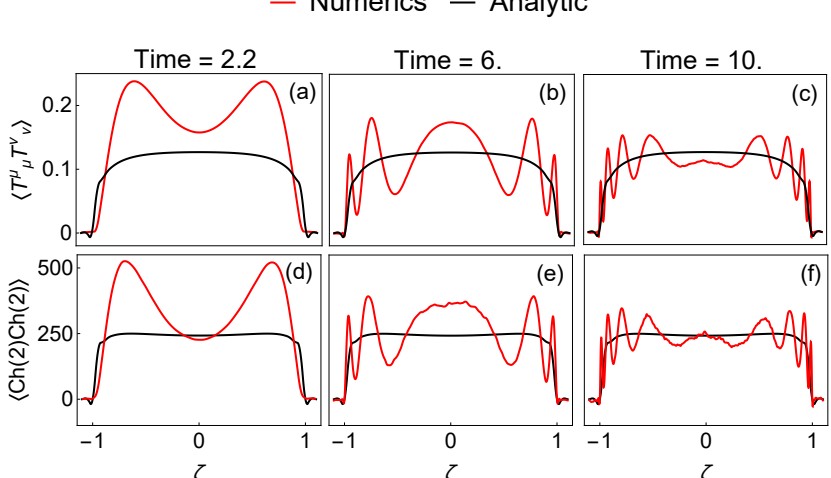

Figure 6: Correlators (41), upper panel (a-c), and (42), lower panel (d-f), are plotted at different times (here and below, we use $\langle \text{Ch(2)Ch(2)} \rangle$ as a shorthand notation for the axis label). Oscillations around the analytic prediction from hydrodynamics are present, although we are unable to conclude if they are asymptotically persisting (with an $O(1)$ amplitude) or not. Parameters: $m = 1$, $g = 2$, inverse temperature $\beta = 1$, the number of samples in the Metropolis is 340000, translational invariance of the system is used as further averaging procedure (500 lattice sites, lattice spacing 0.05). Ripples in the red lines are due to Metropolis noise, enhanced at late times due to the factor of $t$ in front of the correlation function.

and the difference $\langle T^{01}(x,t)T^{01}(0,0)\rangle^c - \langle T^{11}(x,t)T^{11}(0,0)\rangle^c$, and similarly correlation functions involving vertex operator are finite. In the following we focus on two correlation functions: the trace two-point function $\langle T^\mu_\mu(x,t)T^\nu_\nu(0,0)\rangle^c$, and the symmetrized vertex operator two-point function $\langle \cosh(2g\Phi(x,t))\cosh(2g\Phi(0,0))\rangle^c$, because these offer the best chance of numerical precision.

We first remark that, without fluid-cell averaging, one may expect persisting oscillations, of the order of the hydrodynamic value itself, at large space-time variables $(x,t)$. Such oscillations are beyond the validity of the hydrodynamics, which becomes predictive only when fluid cell averages are considered. This is clear at the free-field point $g = 0$: the large-time limit $t \to \infty$ with $x = \zeta t$ and $\zeta$ fixed, of the correlation function rescaled by $t^{-1}$ displays non-vanishing oscillations around the predicted hydrodynamic value, see Appendix B. In the interacting case, the presence of such persisting oscillations is much harder to assess. In Figure 6 we study the scaled correlation functions

$$t\langle T^\mu_\mu(\zeta t, t)T^\nu_\nu(0,0)\rangle^c \tag{41}$$

and

$$t\langle \cosh(2g\Phi(\zeta t, t))\cosh(2g\Phi(0,0))\rangle^c \tag{42}$$

on rays at various times. Relatively large oscillations do arise at all times we were able to reach, although it is inconclusive whether these actually persist asymptotically.

In order to integrate out these oscillations most efficiently, we study the time-averaged and space-integrated scaled correlation functions (24). More precisely, on the one hand time averages along rays directly give the Euler-scale correlation function,

$$\lim_{t \to \infty} \int_0^t \frac{d\tau}{t} \tau \langle \mathcal{O}_1(\zeta\tau, \tau)\mathcal{O}_2(0,0)\rangle^c = \int d\theta \, \delta(\zeta - v^{\text{eff}}(\theta)) \rho_p(\theta) n(\theta) V^{\mathcal{O}_1}(\theta) V^{\mathcal{O}_2}(\theta). \tag{43}$$

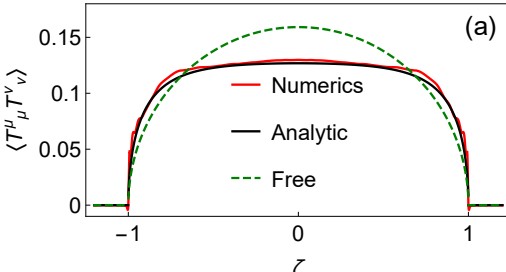
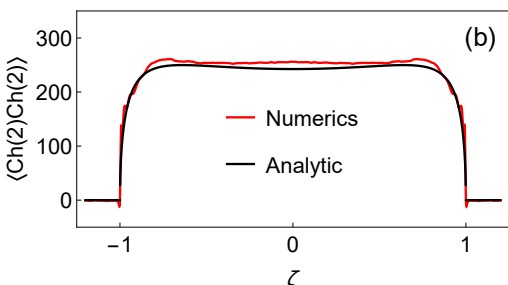

Figure 7: The predictions (black curves) for the time average of two-point functions of (a) the trace of the stress-energy tensor, (44), and (b) vertex operators, (43) with (46), are compared to the numerical simulation (red curves). In the case (a) the free result of Appendix B (dotted curve) is plotted for comparison. In order to improve the convergence, short times are avoided: the time average is taken in the interval $[t_0, t]$, rather than $[0, t]$ as in eq. (44), with $t_0 = 5$ and $t = 20$. The same parameters as those of Figure 6 are used.

We have numerically observed, and analytically calculated in the free case, that it is not necessary, after such time averaging, to perform any mesoscopic space averaging in order to recover the Euler-scale hydrodynamic result: the large time limit as in (43) indeed exists and gives the right-hand side.

On the other hand, space integrations are obtained as (28). In this case, we have analytically observed at the free-field point that, for certain observables, the mesoscopic time averaging $\int_{\mathcal{N}_t(\tau)} \frac{d\tau}{|\mathcal{N}_t|}$ in (28) is indeed *necessary* in order to recover the predicted hydrodynamic result (see Appendix B). That is, at least for correlators in the generic form $\langle e^{kg\Phi} e^{k'g\Phi} \rangle$, undamped time-oscillating terms survive the space integration, but are cancelled by a subsequent fluid-cell time average. In the interacting case, again our numerics does not allow us to reach unambiguous conclusions, although we observe long lived oscillations of the space integrated correlators around the GHD prediction within the numerically accessible time scale. It turns out, however, that mesoscopic time averaging is not necessary for the correlation functions $\langle T_\mu^\mu(x, t) T_\nu^\nu(0, 0) \rangle^c$ and $\langle \cosh(2g\Phi(x, t) \cosh(2g\Phi(0, 0)) \rangle^c$ that we have chosen; this is seen analytically in the free case and numerically observed in the interacting case, and simplifies our numerical checks.

For the trace of the stress-energy tensor, time averaging along rays gives

$$\lim_{t \to \infty} \int_0^t \frac{d\tau}{t} \tau \langle T_\mu^\mu(\zeta\tau, \tau) T_\nu^\nu(0, 0) \rangle^c = m^2 \frac{\rho_p(\theta) n(\theta)}{|\partial_\theta v^{\text{eff}}(\theta)|} \left[ \cosh^{\text{dr}}(\theta) \left( 1 - \left( v^{\text{eff}}(\theta) \right)^2 \right) \right]^2 \bigg|_{v^{\text{eff}}(\theta) = \zeta},$$
(44)

while space integration gives

$$\lim_{t \to \infty} \int dx \, \langle T_\mu^\mu(x, t) T_\nu^\nu(0, 0) \rangle^c = m^2 \int d\theta \, \rho_p(\theta) n(\theta) \left[ \cosh^{\text{dr}}(\theta) \left( 1 - \left( v^{\text{eff}}(\theta) \right)^2 \right) \right]^2. \quad (45)$$

We refer to Appendix B for a direct analytical check of (44) and (45) in the free case, together with an analysis of the leading corrections. For the vertex operators, the hydrodynamic projection functions in (43) are calculated using the GGE one-point averages along with (27). We show in Appendix A that the function $V^{e^{kg\Phi}} = V^{e^{-kg\Phi}} = V^k(\theta)$ is given by

$$V^{k+1}(\theta) = \frac{g^2}{4\pi} \mathcal{V}^{k+1} \sum_{l=0}^k \frac{\mathcal{V}^l}{\mathcal{V}^{l+1}} (2l + 1) n(\theta) \frac{p^l(\theta) d^l(\theta)}{\rho_p(\theta)}, \quad (46)$$

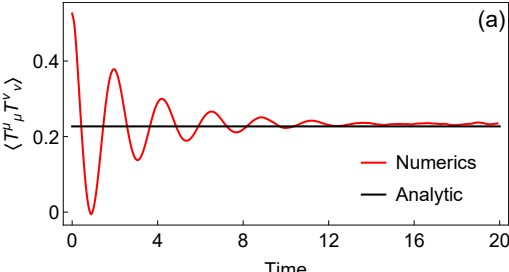
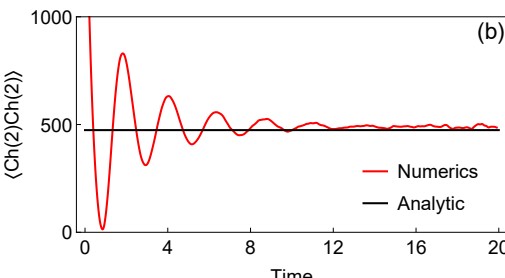

Figure 8: The predictions (black curves) for the space integral of two-point functions of (a) the trace of the stress-energy tensor, (45), and (b) vertex operators, (28) with (46), are compared to the numerical simulation (red curves) as functions of time. At large times, the oscillating ripples are due to the Metropolis noise and are observed to reduce on increasing the number of samples. The same parameters $m, \beta, g$ as those of Figure 6 are used.

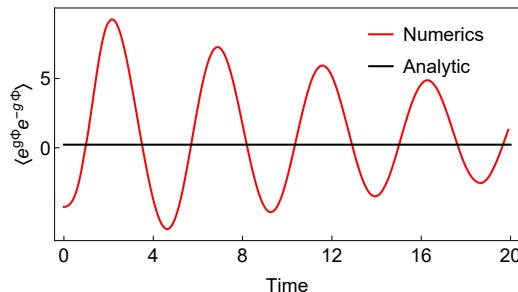

Figure 9: $\int dx \, \langle e^{g\Phi(x,t)} e^{-g\Phi(0,0)} \rangle^c$ is numerically computed (red curve) and compared with the constant predicted by hydrodynamics (black curve). Compared with the correlators of $\cosh(g\Phi)$ (proportional to the trace of the stress-energy tensor analysed in Figure 8), strong oscillations around the GHD prediction on the accessible time scale are observed. The same parameters as those of Figure 8 are used.

where $\mathcal{V}^k = \langle e^{kg\Phi} \rangle$, and where $d^l$ is the solution of the following linear integral equation:

$$d^l(\theta) = e^\theta + \frac{g^2}{4} \mathscr{P} \int \frac{d\gamma}{2\pi} \frac{1}{\sinh(\theta-\gamma)} \left(-2l - \partial_\gamma\right) \left(n(\gamma) d^l(\gamma)\right). \tag{47}$$

Then one directly applies (43) and (28). Again we refer to Appendix B for a direct analytical check of vertex operator correlations functions in the free case.

In Figures 7 and 8 the time-averaged and space-integrated correlators, respectively, are considered. We see very good agreement between the numerical simulation and the hydrodynamic prediction. In Figure 9 we study space-integrated correlation functions of vertex operator without symmetrizing. In this case, as discussed above, we observe long-lived oscillations around the hydrodynamic value.

## 4 Conclusions

In this paper we studied the hydrodynamics of the classical sinh-Gordon model. In the set-ups we studied, the initial state is distributed according to (generalized) Gibbs ensembles, and this distribution is deterministically evolved following the classical field equation. We studied

both one-point averages in the non-equilibrium partitioning protocol, and dynamical two-point functions in homogeneous states. We compared predictions from the new theory of GHD with results from a direct numerical simulation of the protocol. We found excellent agreement in all the cases we studied.

This gives direct evidence that GHD, first developed in the context of quantum chains and quantum field theory and later observed to be applicable to classical integrable soliton-like gases as well, is also applicable to classical field theory, thus further confirming its wide applicability. Because of the UV catastrophe of classical field theory, not all local observables have finite averages in thermal states or GGEs. Nevertheless, we verified that for those whose averages are finite, GHD predictions are correct.

The TBA formulation of GGEs in classical field theory generically involves two types of modes: solitonic modes and radiative modes. The latter are responsible for the UV catastrophe of classical fields. Similarly, the GHD of classical integrable field theory is that for a gas of solitons and radiation. Solitonic modes have a natural classical-particle interpretation, and the part of GHD related to such modes is indeed identical to the GHD of classical soliton gases. Radiative modes, on the other hand, do not have a clear particle interpretation, and were never studied within GHD before. Nevertheless the quasi-particle formulation of GHD still holds, with the appropriate radiative free energy function. In the classical sinh-Gordon model, only radiative modes occur, hence our results give a verification of GHD in this new sector of the theory.

Our results also constitute the first direct verification of the recent GHD correlation function formula. Crucially, we analysed not only conserved fields – specifically the trace of the stress-energy tensor, which is UV finite – but also vertex operators, local fields which are not part of conservation laws. GHD two-point functions for such fields involve the hydrodynamic projection theory, and our results confirm its validity. In addition, our results also provide the first numerical verification of GGE one-point functions formulae for vertex operators recently proposed.

In our study of correlation functions, we observed that GHD predictions are valid up to oscillations which may not subside at large times. We have observed this explicitly in the free-field case, where in certain cases, oscillations both in space and time, of the same order as that of the hydrodynamic predictions, persist indefinitely. For certain observables, appropriate fluid cell averaging appears to be essential in order to recover the Euler-scale correlation predictions from hydrodynamics.

It would be interesting to analyze the approach, at large times, to the GHD predictions. We have explicitly evaluated (and numerically verified) large-time power-law correction terms in the free-field case, but preliminary numerics suggest that in the interacting case, some correction terms are instead exponential. It would also be interesting to extend this analysis to the non-relativistic limit of the sinh-Gordon model, namely the well known nonlinear Schrödinger equation, and to other models, such as the sine-Gordon model, which possesses solitonic modes, and the classical Toda chain.

# Acknowledgments

AB is indebted to Andrea De Luca for useful discussions and to King's College London for hospitality in the framework of the Erasmus Plus traineeship mobility programme. BD is grateful to H. Spohn for discussions, and thanks the Institut d'Études Scientifiques de Cargèse, France and the Perimeter Institute, Waterloo, Canada for hospitality during completion of this work. BD also thanks the Centre for Non-Equilibrium Science (CNES). GMTW would like to thank N. Gromov and P. Xenitidis for discussions on conserved quantities. TY acknowledges support

from the Takenaka Scholarship Foundation.

## A  Semi-classical limit of the sinh-Gordon model

Even though the (generalized) TBA for classical models can be directly accessed by mean of the inverse scattering approach [78], the rather technical derivation appears to be more cumbersome compared with the quantum counterpart, which probably is more familiar to the reader. This appendix provides a self-consistent derivation of the TBA and hydrodynamics for the classical sinh-Gordon, regarding the latter as the classical limit of its quantum version. Concerning the TBA, such a route was already considered in [69], together with the classical limit of the LeClair-Mussardo formula [77] leading to a small excitation-density expansion for one-point functions. Here we resume such a program extending the derivation to the whole hydrodynamics, then we present a close expression for the expectation values of vertex operators obtained through the semi-classical limit of the ansatz presented in [69]. While in agreement with the LeClair-Mussardo formula, this last result is not constrained to small excitation densities and is feasible to be applied to arbitrary GGEs.

The quantum sinh-Gordon is arguably one of the simplest, yet interacting, quantum field theories. Its spectrum consists of only one type of particle [75], leading to a simpler version of the TBA equation (4)

$$\epsilon_{\mathrm{q}}(\theta) = w(\theta) - \int \frac{\mathrm{d}\gamma}{2\pi} \, \varphi_{\mathrm{q}}(\theta - \gamma) \log\left(1 + e^{-\epsilon_{\mathrm{q}}(\gamma)}\right), \tag{48}$$

where we explicitly use the subscript "q" as a remind we are dealing now with the *quantum* model; in the absence of any subscript we will refer to classical quantities. Differently from [69], along the semi-classical limit procedure we will use the fermionic formulation of the quantum ShG rather than the bosonic one: even though the two descriptions are equivalent [69], the quantum hydrodynamics is formulated in the fermionic basis making the latter a preferred choice. Referring to the parameter choice of the (quantum) Lagrangian (29), the quantum kernel $\varphi_{\mathrm{q}}$ is

$$\varphi_{\mathrm{q}}(\theta) = \frac{2\sin\pi\alpha}{\sinh^2\theta + \sin^2\pi\alpha}, \qquad \alpha = \frac{g^2}{8\pi + g^2}. \tag{49}$$

The physical mass of the particles $m_{\mathrm{q}}$ is related to the bare mass $m$ through the relation [79]

$$m^2 = m_{\mathrm{q}}^2 \frac{\pi\alpha}{\sin\pi\alpha}. \tag{50}$$

As already pointed out in [69], the semi-classical limit of the ShG model can be obtained through a combination of a small coupling and high temperature limit. More precisely we consider the rescaling

$$\beta_i \to \hbar\beta_i, \qquad g \to \sqrt{\hbar}g, \qquad \mathcal{O}[\Phi] \to \mathcal{O}[\sqrt{\hbar}\Phi], \tag{51}$$

with $\mathcal{O}$ a generic observable functional of the field and $\beta_i$ the Lagrange multipliers of the GGE. Then, in the $\hbar \to 0$ limit, the quantum expectation value reduces to the expectation value of the classical observable $\mathcal{O}[\Phi]$ on a GGE having $\beta_i$ as Lagrange multipliers. Such a claim is easily checked in the simplest case of a thermal state: using the path integral representation of the latter we have

$$\langle \mathcal{O}(\sqrt{\hbar}\Phi) \rangle_{\mathrm{q}}^{\hbar\beta} = \frac{1}{\mathcal{Z}} \int \mathcal{D}\Phi \, \mathcal{O}(\sqrt{\hbar}\Phi) \, e^{-\int_0^{\hbar\beta} \mathrm{d}\tau \int \mathrm{d}x \frac{1}{2}\partial_\mu\Phi\partial^\mu\Phi + \frac{m^2}{g^2\hbar}(\cosh(\sqrt{\hbar}g\Phi)-1)}, \tag{52}$$

where the euclidean time $\tau$ runs on a ring of length $\hbar\beta$. In the high temperature limit the integral over the compact dimension can be approximated with the integrand at $\tau = 0$ times the length of the ring. In the same limit the measure collapses to $\mathscr{D}\Phi\,\mathscr{D}\partial_\tau\Phi$ where the fields are now restricted to $\tau = 0$,

$$\langle \mathcal{O}(\sqrt{\hbar}\Phi)\rangle_{\mathrm{q}}^{\hbar\beta} \simeq \frac{1}{\mathscr{Z}} \int \mathscr{D}\Phi\,\mathscr{D}\partial_\tau\Phi \;\; \mathcal{O}(\sqrt{\hbar}\Phi)\,e^{-\hbar\beta\int \mathrm{d}x\,\frac{1}{2}\partial_\mu\Phi\partial^\mu\Phi + \frac{m^2}{g^2\hbar}(\cosh(\sqrt{\hbar}g\Phi)-1)}. \tag{53}$$

A final change of variable in the path integral $\Phi \to \Phi/\sqrt{\hbar}$ makes explicit the appearance of the classical expectation value of the observable $\mathcal{O}$ on a classical thermal ensemble of inverse temperature $\beta$. A similar reasoning can be extended to more general GGEs as well to the dynamics, confirming (51) as the correct scaling to obtain the desired semi-classical limit. Notice that, in such a limit, the physical quantum mass simply reduces to the bare mass. Concerning the TBA equation (48), the rescaling of the Lagrange multipliers is simply translated in $\omega(\theta) \to \hbar\omega(\theta)$. A finite integral equation in the $\hbar \to 0$ limit is obtained provided we redefine $\epsilon_{\mathrm{q}}(\theta) = \log\epsilon(\theta) + \log\hbar$, with $\epsilon$ the classical effective energy (this rescaling differs from that of [69], where the *bosonic* rather than the *fermionic* quantum effective energy is used). Through straightforward manipulations, at the leading order in $\hbar$ the following integral equation is obtained

$$\epsilon(\theta) = \omega(\theta) + \frac{g^2}{4}\lim_{\hbar\to 0}\int \frac{d\theta'}{2\pi}\frac{\cosh(\theta - \theta')}{\sinh^2(\theta - \theta') + (\hbar g^2/8)^2}\Big(\log\epsilon(\theta') - \log\epsilon(\theta)\Big). \tag{54}$$

The last step involves an integration by parts that leads to the final $\hbar-$independent integral equation

$$\epsilon(\theta) = \omega(\theta) - \frac{g^2}{4}\mathscr{P}\int \frac{\mathrm{d}\gamma}{2\pi}\frac{1}{\sinh(\theta - \gamma)}\partial_\gamma\log\epsilon(\gamma), \tag{55}$$

where $\mathscr{P}$ stands for the principal value prescription, natural emerging passing from (54) to (55): this is the sought-after TBA equation for the classical ShG. We should warn the reader about a difference in the sign in front of the integral between the above expression and the same reported in Ref. [69], where a typo must have occurred. This concludes the pure TBA calculation and we can now pass to the hydrodynamics: rather than taking the differential scattering phase for the classical ShG from (55) and plug it into the hydrodynamic formulas of Section 2.2, we directly take the limit of the GHD for quantum integrable systems as a further check of the correctness of its classical version. Through calculations close to those that provided the classical TBA, it is possible to show that the quantum dressing operation (14) (here specialized to the ShG case)

$$h_{\mathrm{q}}^{\mathrm{dr}}(\theta) = h(\theta) + \int \frac{\mathrm{d}\gamma}{2\pi}\varphi_q(\theta - \gamma)n_{\mathrm{q}}(\gamma)h_{\mathrm{q}}^{\mathrm{dr}}(\gamma) \tag{56}$$

reduces to the sought definition of the classical dressing

$$h^{\mathrm{dr}}(\theta) = h(\theta) - \frac{g^2}{4}\mathscr{P}\int \frac{\mathrm{d}\gamma}{2\pi}\frac{1}{\sinh(\theta - \gamma)}\partial_\gamma[n(\gamma)h^{\mathrm{dr}}(\gamma)], \tag{57}$$

provided we set $h_{\mathrm{q}}^{\mathrm{dr}}(\theta) = \hbar^{-1}n(\theta)h^{\mathrm{dr}}(\theta)$ while taking the $\hbar \to 0$ limit. In particular, this implies that the effective velocity (16) of the quantum model converges, in the semi-classical limit, to the definition of the effective velocity in the classical case. Finally, from the definition of the occupation function $n(\theta)$ in the classical (13) and quantum (17) cases and the rescaling of the effective energies we have, at the leading order in $\hbar$, $n_{\mathrm{q}}(\theta) = 1 - \hbar/n(\theta) + ...$ . Replacing this last piece of information in the quantum version of the GHD (21) we readily obtain its

classical counterpart. Alternatively, we could have noticed that in the semi-classical limit the expectation values of the quantum conserved charges and their currents go, respectively, to the classical expectation values of charges and currents as written in eq. (10-11) (apart from an overall and inessential $\hbar$ factor) and impose the conservation laws.

### A.1 The expectation value of the vertex operators

This part is devoted to the derivation of the equations (38-39) reported in the main text, through the semi-classical limit of the ansatz presented in Ref. [55]. Following the latter, the expectation values of vertex operators in a GGE for the quantum model satisfy

$$\frac{\langle e^{(k+1)g\Phi}\rangle_{\mathrm{q}}}{\langle e^{kg\Phi}\rangle_{\mathrm{q}}} = 1 + \frac{2\sin(\pi\alpha(2k+1))}{\pi}\int \mathrm{d}\theta\, \frac{e^{\theta}}{1+e^{\epsilon_{\mathrm{q}}(\theta)}}p_{\mathrm{q}}^{k}(\theta), \tag{58}$$

with

$$p_{\mathrm{q}}^{k}(\theta) = e^{-\theta} + \int \mathrm{d}\gamma\, \frac{1}{1+e^{\epsilon_{\mathrm{q}}(\mu)}}\chi_{k}(\theta-\gamma)p_{\mathrm{q}}^{k}(\gamma) \tag{59}$$

$$\chi_{k}(\theta) = \frac{i}{2\pi}\left(\frac{e^{-i2k\alpha\pi}}{\sinh(\theta+i\pi\alpha)} - \frac{e^{i2k\alpha\pi}}{\sinh(\theta-i\pi\alpha)}\right). \tag{60}$$

As it is evident from the rescaling (51), the quantum expectation value of the vertex operators goes, in the semi-classical limit, directly to the classical expectation value of the latter (the rescaling of the coupling $g$ and of the field in the observable cancel each others). Thus the classical limit of the r.h.s. of (58) will directly produce the ratio of the expectation values of the vertex operators in the classical theory. The first step to take the limit is to consider $\chi_k$ up to first order in $\hbar$. Using that $\alpha$ after the rescaling is first order in $\hbar$, we eventually need (in the distribution sense)

$$\frac{1}{\sinh(\theta-\gamma-i\pi\alpha)} = \tag{61}$$
$$i\pi\delta(\theta-\gamma) + \mathscr{P}\frac{1}{\sinh(\theta-\gamma)} + \pi\alpha\left(\pi\delta(\theta-\gamma)\partial_{\theta} - i\mathscr{P}\frac{1}{\sinh(\theta-\gamma)}\partial_{\gamma}\right) + \mathscr{O}(\alpha^{2}).$$

In order to get a finite expression in the $\hbar \to 0$ limit, we define the classical $p^k$ function as $p_{\mathrm{q}}^{k}(\theta) = \hbar^{-1}n(\theta)p^{k}(\theta)$. Using this definition and (61) is a matter of few straightforward passages to obtain the finite classical integral equations (38-39).

### A.2 The Hydrodynamic projection of the vertex operators

Having access to the exact value of the expectation value of vertex operators on arbitrary GGEs makes the exact computation of the hydrodynamic projection (27) feasible, paving the way to correlation functions on an Euler scale. Specializing (27) to the ShG model and to the vertex operators, we are interested in computing

$$-\frac{\partial \mathscr{V}^{k}}{\partial \beta_{j}} = \int \mathrm{d}\theta\rho_{\mathbf{p}}(\theta)n(\theta)h_{j}^{\mathrm{dr}}(\theta)V^{k}(\theta), \tag{62}$$

where for simplicity $\mathscr{V}^{k} = \langle e^{kg\phi}\rangle$ and $\beta_j$ is the Lagrange multiplier associated with the charge whose eigenvalue is $h_j$. Differentiating both sides of eq. (38) with respect to $\beta_j$, we readily obtain the recurrence relation

$$\frac{\delta \mathscr{V}^{k+1}}{\mathscr{V}^{k+1}} = \frac{\delta \mathscr{V}^{k}}{\mathscr{V}^{k}} + \frac{\mathscr{V}^{k}}{\mathscr{V}^{k+1}}(2k+1)\frac{g^{2}}{4\pi}\int \mathrm{d}\theta\, e^{\theta}\partial_{\beta_{j}}(n(\theta)p^{k}(\theta)). \tag{63}$$

That has the obvious solution

$$\frac{\delta \mathcal{V}^{k+1}}{\mathcal{V}^{k+1}} = \sum_{l=0}^{k} \frac{\mathcal{V}^l}{\mathcal{V}^{l+1}} (2l+1) \frac{g^2}{4\pi} \int d\theta \, e^{\theta} \partial_{\beta_j} (n(\theta) p^l(\theta)) \,. \tag{64}$$

$\partial_{\beta_j}(np^l)$ can now be extracted from the integral equation (39). To carry on the calculation, it is convenient to introduce an operatorial representation of the integration kernel. In particular, we define the linear operators $\hat{n}$ and $\hat{\varphi}_l$ as follows:

$$(\hat{n}\tau)(\theta) = n(\theta)\tau(\theta), \qquad (\hat{\varphi}_l \tau)(\theta) = \int d\gamma \frac{g^2}{4} \frac{1}{\sinh(\theta - \gamma)} (\partial_\gamma - 2l)\tau(\gamma) \,, \tag{65}$$

for any given test function $\tau(\theta)$. In this language, the linear equation satisfied by $p^l$ (39) is written as

$$p^l = e^- - \frac{1}{2\pi} \hat{\varphi}_l \hat{n} p^l \,, \tag{66}$$

where we introduce $(e^-)(\theta) = e^{-\theta}$. The formal solution is immediately written

$$\hat{n} p^l = (\hat{n}^{-1} + \frac{1}{2\pi} \hat{\varphi}_l)^{-1} e^- \,, \tag{67}$$

and the derivative with respect to $\beta_j$ is easily performed, using the fact that $n(\theta) = 1/\epsilon(\theta)$ and the TBA equations (31)

$$\partial_j(\hat{n} p^l) = -(\hat{n}^{-1} + \frac{1}{2\pi} \hat{\varphi}_l)^{-1} \hat{n} p^l h_j^{\text{dr}} \,. \tag{68}$$

As last technical step, we consider the integral appearing in eq. (64)

$$\int d\theta \, e^{\theta} \partial_j(n(\theta) p^l(\theta)) = -\int d\theta d\gamma \, e^{\theta} (\hat{n}^{-1} + \frac{1}{2\pi} \hat{\varphi}_l)^{-1}_{(\theta,\gamma)} n(\gamma) p^l(\gamma) h_j^{\text{dr}}(\gamma) \,, \tag{69}$$

where we made explicit the convolution underlying the action of the operator. We define $d^l$ as

$$n(\gamma) d^l(\gamma) = \int d\theta \, e^{\theta} (\hat{n}^{-1} + \frac{1}{2\pi} \hat{\varphi}_l)^{-1}_{(\theta,\gamma)} \,. \tag{70}$$

Using the symmetry $(\hat{n}^{-1} + \frac{1}{2\pi}\hat{\varphi}_l)^{-1}_{(\theta,\gamma)} = (\hat{n}^{-1} + \frac{1}{2\pi}\hat{\varphi}_{-l})^{-1}_{(\gamma,\theta)}$ it is immediate to see that $d^l$ satisfies the linear integral equation

$$d^l(\theta) = e^{\theta} + \frac{g^2}{4} \mathcal{P} \int \frac{d\gamma}{2\pi} \frac{1}{\sinh(\theta - \gamma)} (-2l - \partial_\gamma)(n(\gamma) d^l(\gamma)) \,, \tag{71}$$

i.e. eq. (47) of Section 3.3. Using $d^l$ in eq. (69), plugging the latter in (64) and finally comparing it with eq. (62), we finally take the desired result

$$V^{k+1}(\theta) = \frac{g^2}{4\pi} \mathcal{V}^{k+1} \sum_{l=0}^{k} \frac{\mathcal{V}^l}{\mathcal{V}^{l+1}} (2l+1) n(\theta) \frac{p^l(\theta) d^l(\theta)}{\rho_{\mathbf{p}}(\theta)} \,, \tag{72}$$

i.e. eq. (46) of Section 3.3.

# B  The free field limit

In this section we focus on the free limit of the sinh-Gordon model (equivalent to the zero temperature limit), i.e. the free boson model. In particular, we consider the profile of the stress energy tensor as well as vertex operators in the partitioning protocol presented in Section 3.2. Within an homogeneous ensemble, we study the relevant correlators $\langle T^\mu_\mu(x,t) T^\nu_\nu(0,0)\rangle^c$ and $\langle e^{kg\Phi(x,t)} e^{k'g\Phi(0,0)}\rangle^c$, in the suitable Eulerian limit, providing a check of the content of Section 3.3 in the free field limit.

## B.1  Preliminaries

We first recall some elementary properties of the free model. Its Hamiltonian is given by

$$H = \frac{1}{2}\int \mathrm{d}x((\partial_t\Phi)^2 + (\partial_x\Phi)^2 + m^2\Phi^2), \tag{73}$$

and the field $\Phi(x,t)$ is mode expanded as

$$\Phi(x,t) = \int \frac{\mathrm{d}p}{2\pi}\frac{1}{\sqrt{2E(p)}}\big[A(p)e^{-iE(p)t+ipx} + \bar{A}(p)e^{iE(p)t-ipx}\big], \tag{74}$$

where, rather than reasoning in terms of rapidities, we find it more convenient to consider the momentum $p = m\sinh\theta$. Above, $E(p) = \sqrt{p^2+m^2}$ is of course the energy of the mode. The modes $A(p)$ diagonalize the Hamiltonian

$$H = \int \frac{\mathrm{d}p}{2\pi}E(p)|A(p)|^2, \tag{75}$$

as well all the other local charges the free mode possesses. In this perspective, probability densities associated with GGE-like ensembles acquire a remarkably simple appearance

$$\rho_{\mathrm{GGE}} = \frac{e^{-\int \frac{\mathrm{d}p}{2\pi}\omega(p)|A(p)|^2}}{\mathscr{Z}}. \tag{76}$$

Free GGEs are therefore *gaussian* in the modes $A(p)$ (and this implies gaussianity in $\Phi$ as well), with zero mean $\langle A(p)\rangle = 0$ and correlation $\langle A(p)\bar{A}(q)\rangle = 2\pi\delta(p-q)n(p)$ with $n(p) = 1/\omega(p)$. On thermal ensembles, we simply have $\omega(p) = \beta E(p)$, in agreement with the free field limit of the TBA equations.

## B.2  The partitioning protocol

Within the framework of the free theory, the partitioning protocol of Section 3.2 is most easily solved. In fact, as no dressing is present, eq. (23) is entirely expressed in terms of explicit quantities, with the group velocity in place of the effective velocity. In the Klein-Gordon case (and using momenta rather then rapidities) it reads

$$n(\zeta;p) = n_L(p)\Theta\big(p - m\zeta(1-\zeta^2)^{-1/2}\big) + n_R(p)\Theta\big(m\zeta(1-\zeta^2)^{-1/2} - p\big), \tag{77}$$

where we inverted the group velocity function

$$v^{\mathrm{gr}}(p) = \frac{p}{\sqrt{p^2+m^2}}. \tag{78}$$

In the spirit of GHD, the profile of a given observable $\mathcal{O}(x, t)$ is simply computed evaluating $\langle \mathcal{O}(x, t) \rangle$ in an homogeneous GGE, but where the inhomogeneous filling $n(\zeta; p)$ must be employed. We can now consider the profile of the trace of the stress energy tensor, which in the free case simply reduces to $T^\mu_\mu(x, t) = m^2 \Phi^2(x, t)$. Then according to hydrodynamics,

$$\lim_{t \to \infty} \langle T^\mu_\mu(t\zeta, t) \rangle = m^2 \int \frac{dp}{2\pi} \frac{n(\zeta; p)}{E(p)} . \tag{79}$$

In particular, if we consider an initially bipartite thermal ensemble of inverse temperatures $\beta_L$ and $\beta_R$ respectively, the integral can be computed exactly giving

$$\langle T^\mu_\mu \rangle = \frac{m}{4} \left( \frac{1}{\beta_L} + \frac{1}{\beta_R} \right) + \frac{m}{2\pi} \left( \frac{1}{\beta_R} - \frac{1}{\beta_L} \right) \tan^{-1} \left[ \frac{\zeta}{\sqrt{1 - \zeta^2}} \right] . \tag{80}$$

The profiles of the vertex operators are easily computable as well. Of course, in order to obtain a non-trivial result while $g \to 0$ we should rescale $k$ in the vertex operator $e^{kg\Phi}$ in such a way that $kg$ remains constant. Vertex operators are most easily addressed exploiting the gaussianity of the ensemble through the cumulant expansion

$$\langle e^{kg\Phi(x,t)} \rangle = e^{\frac{(kg)^2}{2} \langle \Phi^2(x,t) \rangle} \tag{81}$$

and $\Phi^2(x, t)$, apart from an overall $m^2$ factor, is nothing other than the trace of the stress energy tensor we considered before.

We note that as soon as the interaction is switched on, the simple form (80) is spoiled by the dressing of the energy and of the effective velocity, nevertheless a trait of the free theory survives to the dressing procedure, namely the behaviour at the edges of the light-cone $\zeta = \pm 1$. In fact, eq. (80) approaches the edges with a square root singularity in the first derivative and such a behaviour is clearly present also in the interacting plots of Section 3.2. The motivation is simple: the behaviour at the light-cone is determined only by the fastest quasi-particles, but for large rapidities the filling $n(\theta)$ goes to zero and the dressing becomes ineffective. In this limit the theory is essentially free, recovering thus the behaviour of (80).

### B.3   Correlation function of the stress-energy tensor $T^\mu_\mu(x, t)$

Having reviewed the basics of the free theory, we are now in the position to compute the connected correlator

$$\langle T^\mu_\mu(x, t) T^\nu_\nu(0, 0) \rangle^c = m^4 \langle \Phi(x, t)^2 \Phi(0, 0)^2 \rangle^c = 2m^4 \left( \langle \Phi(x, t) \Phi(0, 0) \rangle \right)^2 . \tag{82}$$

where in the last step we took advantage of gaussianity of the GGE ensemble. A quick evaluation of the two point correlator leads us to

$$\langle T^\mu_\mu(x, t) T^\nu_\nu(0, 0) \rangle^c = 2m^4 \left( \int \frac{dp}{2\pi} \frac{n(p)}{E(p)} \cos(E(p)t - px) \right)^2 . \tag{83}$$

We can now consider the two relevant Eulerian scale limits analysed in the main text. First, we study the two point correlator averaged along the ray, i.e.

$$\lim_{t \to \infty} \frac{1}{t} \int_0^t d\tau \, \tau \langle T^\mu_\mu(\tau\zeta, \tau) T^\nu_\nu(0, 0) \rangle^c . \tag{84}$$

In the large $t$ limit, only the large time limit of the correlator matters. In fact, any integration on a finite region $0 < \tau < t_0$ gives $\sim t^{-1}$ corrections. The large time limit is immediately computed by mean of a saddle point approximation. We define $p_\zeta$ as the solution of $v^{gr}(p_\zeta) = \zeta$. Then

$$\int \frac{dp}{2\pi} \frac{n(p)}{E(p)} \cos(E(p)t - px) = \frac{n(p_\zeta)}{2\pi E(p_\zeta)} \cos(t(E(p_\zeta) - \zeta p_\zeta) + \pi/4) \sqrt{\frac{2\pi}{t|\partial_{p_\zeta} v(p_\zeta)|}} + \mathcal{O}(t^{-1}). \tag{85}$$

Plugging this in (83) we readily obtain

$$\frac{1}{t} \int_0^t d\tau\, \tau \langle T^\mu_\mu(\tau\zeta, \tau) T^\nu_\nu(0,0)\rangle^c = \frac{m^4}{2\pi} \left(\frac{n(p_\zeta)}{E(p_\zeta)}\right)^2 \frac{1}{|\partial_{p_\zeta} v(p_\zeta)|} + \mathcal{O}(t^{-1}). \tag{86}$$

This coincides with the GHD prediction in the non-interacting case (44). In the case of a free thermal ensemble we get

$$\frac{1}{t} \int_0^t d\tau\, \tau \langle T^\mu_\mu(\tau\zeta, \tau) T^\nu_\nu(0,0)\rangle^c = \frac{m}{2\pi\beta^2} \sqrt{1-\zeta^2} + \mathcal{O}(t^{-1}). \tag{87}$$

We can now consider the second Eulerian correlator integrated on space

$$\int dx \langle T^\mu_\mu(x,t) T^\nu_\nu(0,0)\rangle^c = m^4 \int \frac{dp}{2\pi} \left(\frac{n(p)}{E(p)}\right)^2 [1 + \cos(2E(p)t)]. \tag{88}$$

In the long time limit, the fast oscillating phase averages to zero. The leading correction can be computed by mean of a saddle point estimation, giving a $\sim t^{-1/2}$ contribution

$$\int dx \langle T^\mu_\mu(x,t) T^\nu_\nu(0,0)\rangle^c = m^4 \int \frac{dp}{2\pi} \left(\frac{n(p)}{E(p)}\right)^2 + \mathcal{O}(t^{-1/2}), \tag{89}$$

in agreement with the GHD prediction without the need of any further fluid-cell integration along the time direction.

## B.4 Correlation function of the vertex operators

Besides the correlation functions of the trace of the stress energy tensor, the GHD provides us the value of correlation functions of the vertex operators as well. This part is devoted to computing $\langle e^{kg\Phi(x,t)} e^{k'g\Phi(0,0)}\rangle^c$ on free GGE-like ensembles. Exploiting the gaussianity of the ensemble, we can readily write

$$\langle e^{kg\Phi(x,t)} e^{k'g\Phi(0,0)}\rangle^c = e^{\frac{(kg)^2}{2}\langle\Phi^2(x,t)\rangle} e^{\frac{(k'g)^2}{2}\langle\Phi^2(0,0)\rangle} \left(e^{kk'g^2\langle\Phi(x,t)\Phi(0,0)\rangle} - 1\right). \tag{90}$$

Thanks to the homogeneity of the ensemble, all the non-trivial coordinate and time dependence is carried by $e^{kk'g^2\langle\Phi(x,t)\Phi(0,0)\rangle}$. We start with the correlation function integrated along the ray, in this perspective we are interested in

$$\frac{1}{t} \int_0^t d\tau\, \tau \left(e^{kk'g^2\langle\Phi(\zeta\tau,\tau)\Phi(0,0)\rangle} - 1\right) \tag{91}$$

in the $t \to \infty$ limit. As in the previous case, the leading behaviour can be extracted focusing in the large $\tau$ behaviour of the integrand. In this perspective, $\langle\Phi(\zeta\tau, \tau)\Phi(0,0)\rangle$ can be estimated through the saddle-point as in (85). Then, since $\langle\Phi(\zeta\tau, \tau)\Phi(0,0)\rangle \to 0$ in the $\tau \to \infty$ limit,

we can Taylor expand the exponential. In order to compute the leading behaviour, we need to retain up to the second non-trivial term in the Taylor expansion

$$\frac{1}{t}\int_0^t \mathrm{d}\tau\, \tau\left(e^{kk'g^2\langle\Phi(\zeta\tau,\tau)\Phi(0,0)\rangle}-1\right) = kk'g^2\frac{1}{t}\int_0^t \mathrm{d}\tau\, \tau\langle\Phi(\zeta\tau,\tau)\Phi(0,0)\rangle + \dots$$

$$\dots \frac{(kk'g^2)^2}{2}\frac{1}{t}\int_0^t \mathrm{d}\tau\, \tau\Big(\langle\Phi(\zeta\tau,\tau)\Phi(0,0)\rangle\Big)^2 + \mathcal{O}(t^{-1}). \tag{92}$$

Among the two terms, the second is the same we encountered in the analogue calculations for the $\langle T^\mu_{\ \mu} T^\nu_{\ \nu}\rangle$ correlator and thus attains a constant value plus $\mathcal{O}(t^{-1})$ corrections. Concerning the first term, we are going to show

$$\frac{1}{t}\int_0^t \mathrm{d}\tau\, \tau\langle\Phi(\zeta\tau,\tau)\Phi(0,0)\rangle = \frac{1}{t}\int_0^t \mathrm{d}\tau\, \tau\int\frac{\mathrm{d}p}{2\pi}\frac{n(p)}{E(p)}\cos(\tau(E(p)-p\zeta)) = \mathcal{O}(t^{-1/2}). \tag{93}$$

In fact, exchanging the integration over momentum and time we face

$$\int\frac{\mathrm{d}p}{2\pi}\frac{n(p)}{E(p)}\left[\frac{\sin(t(E(p)-p\zeta))}{E(p)-p\zeta} + \frac{1}{t}\frac{\cos(t(E(p)-p\zeta))-1}{(E(p)-p\zeta)^2}\right] \tag{94}$$

and then a saddle point estimation gives (93). Merging the different pieces, we find

$$\frac{1}{t}\int_0^t \mathrm{d}\tau\, \tau\langle e^{kg\Phi(\tau\zeta,\tau)}e^{k'g\Phi(0,0)}\rangle^c = \left(kk'mg^2\frac{n(p_\zeta)}{E(p_\zeta)}\right)^2\frac{e^{g^2\frac{k^2+k'^2}{2}\langle\Phi^2\rangle}}{8\pi|\partial_{p_\zeta}v(p_\zeta)|} + kk'\mathcal{O}(t^{-1/2}). \tag{95}$$

The consistency of this result with the GHD prediction (28) is easily proven taking the $g\to 0$ limit (with $kg$ constant) of eq. (46) which simply reads

$$\lim_{g\to 0,\, kg\ \mathrm{const.}} V^k(\theta) = \frac{(gk)^2}{2E(m\sinh\theta)}\mathcal{V}^k \tag{96}$$

and then using this in eq. (28). Compared with the stress energy tensor case, the corrections vanish slower (as $\sim t^{-1/2}$ rather then $\sim t^{-1}$) and this qualitative feature can be recognized also in the numerical analysis of the interacting case. However, notice that the $\sim t^{-1/2}$ corrections are proportional to $kk'$. Thus, if we symmetrize the correlator with respect to $k\to -k$ and/or $k'\to -k'$ we can improve the convergence. For example

$$\frac{1}{t}\int_0^t \mathrm{d}\tau\, \tau\langle\cosh(kg\Phi(\tau\zeta,\tau))\cosh(k'g\Phi(0,0))\rangle^c = \left(kk'g^2\frac{n(p_\zeta)}{E(p_\zeta)}\right)^2\frac{e^{g^2\frac{k^2+k'^2}{2}\langle\Phi^2\rangle}}{8\pi|\partial_{p_\zeta}v(p_\zeta)|} + \mathcal{O}(t^{-1}). \tag{97}$$

Also in the interacting case, while we cannot analytically derive the corrections to GHD, we experience a steady improvement in the convergence through symmetrization of the correlators of vertex operators.

We now consider the space-integrated correlator, best addressed by Taylor expanding the exponential

$$\int \mathrm{d}x\left(e^{kk'g^2\langle\Phi(x,t)\Phi(0,0)\rangle}-1\right) = \sum_{j=1}^\infty \frac{(kk'g^2)^j}{j!}\int \mathrm{d}x\int\frac{\mathrm{d}^j p}{(2\pi)^j}\prod_{i=1}^j\left[\frac{n(p_i)}{E(p_i)}\cos(tE(p_i)-p_i x))\right]. \tag{98}$$

Thanks to the space integration, in each term we obtain a global conservation law on the total momentum.

$$
\int dx \left( e^{kk'g^2 \langle \Phi(x,t)\Phi(0,0)\rangle} - 1 \right) =
$$

$$
\sum_{j=1}^{\infty} \frac{(kk'g^2)^j}{j!} \sum_{\pm_i} \int \frac{d^j p}{(2\pi)^j} \prod_{i=1}^{j} \left[ \frac{n(p_i)}{2E(p_i)} \right] e^{it \sum_{i=1}^{j} \pm_i E(p_i)} 2\pi \delta \left( \sum_{i=1}^{n} \pm_i p_i \right). \tag{99}
$$

Above, the sum over all the possible choices of $\pm_i$ must be considered. Now we can study the $t \to \infty$ limit. For $j > 2$ the decay can be readily estimated through a saddle point, leading to the general decay $\sim t^{-(j-1)/2}$. Concerning $j = 2$, this estimation does not hold: in fact there are terms that do not oscillate, plus oscillating phases that provide $t^{-1/2}$ corrections. Crucial is the role of the $j = 1$ term that does not decay, but rather oscillates without any damping.

$$
\int dx \left( e^{kk'g^2 \langle \Phi(x,t)\Phi(0,0)\rangle} - 1 \right) = (kk'g^2) \frac{n(0)}{E(0)} \cos(mt) + \frac{(kk'g^2)^2}{4} \int \frac{dp}{2\pi} \left( \frac{n(p)}{E(p)} \right)^2 + \mathcal{O}(t^{-1/2}). \tag{100}
$$

The presence of the undamped oscillating term has as a consequence that the mere spatial integration is not enough to ensure convergence to GHD and a proper fluid-cell average in the time direction must be considered. However, notice that if we consider the symmetrized correlator $\langle \cosh(kg\Phi) \cosh(k'g\Phi) \rangle$, the undamped term drops out by the symmetry $k \to -k$ or equivalently $k' \to -k'$. In this case, spatial integration alone ensures the convergence to a steady value that matches the non-interacting limit of the GHD prediction. In the interacting case, the numerics displays long-lived oscillations in the space-integrated correlator of general vertex operators, while these are absent in the symmetric case (see Section 3.3).

## C  Numerical Methods

This Appendix contains a short summary of the numerical methods we used to find the results for the Sinh Gordon model presented in Section 3. In Section C.1 we consider the numerical solution of the TBA and hydrodynamics, while in Section C.2 we present the direct simulation of the model.

### C.1  Numerical solution of TBA and hydrodynamic

Both in the partitioning protocol of Section 3.2 and for the correlation function of Section 3.3, the first step is solving the TBA equation (31), which is more difficult than the more familiar quantum case. The main difficulties reside in *i)* the singular kernel (32) and *ii)* the lack of convergence of (31) under a natural iterative approximation scheme, in contrast with the familiar quantum case. The first issue is overcome with a careful discretization of the integral equation, while the second issue is avoided using the Newton method. Consider the integral equation (31) where we set $m = g = 1$ for simplicity and specialize it to the thermal case $\omega(\theta) = \beta \cosh \theta$. Actually, it is convenient to parametrize the effective energy as

$$
\epsilon(\theta) = \beta \cosh \theta \, e^{\chi(\theta)}. \tag{101}
$$

This ensures the UV behaviour $\lim_{\theta \to \pm\infty} \chi(\theta) = 0$. In terms of the new variable eq. (31) can be written as

$$
e^{\chi(\theta)} - 1 + f(\theta) + \frac{1}{4\beta \cosh \theta} \mathcal{P} \int_{-\infty}^{\infty} \frac{d\theta'}{2\pi} \frac{1}{\sinh(\theta - \gamma)} \partial_\gamma \chi(\gamma) = 0, \tag{102}
$$

where the principal value prescription is enforced and $f(\theta)$ is defined as

$$f(\theta) = \frac{1}{4\beta \cosh \theta} \mathscr{P} \int_{-\infty}^{\infty} \frac{d\gamma}{2\pi} \frac{\tanh \gamma}{\sinh(\theta - \gamma)} \,. \tag{103}$$

In order to solve for $\chi(\theta)$ we symmetrically discretize the set of rapidities $\theta_i = \Delta(i - 1/2)$ with $i \in \{-N + 1, -N, ..., N\}$. The linear operator defined through the integral is discretized as follows: defining $t(\theta) = \partial_\theta \chi(\theta)$ we pose

$$\int_{-\infty}^{\infty} d\gamma \frac{t(\gamma)}{\sinh(\theta_i - \gamma)} \simeq \sum_{j=-N+1}^{N} \int_{\theta_j - \Delta/2}^{\theta_j + \Delta/2} d\gamma \frac{\sum_{a=0}^{2l} A_i^a[t](\gamma - \theta_j)^{2a}}{\sinh(\theta_i - \gamma)} \,, \tag{104}$$

where in each interval centred in $\theta_j$ we replace $t(\theta)$ with a symmetrical interpolation of the $2l^{\text{th}}$ order $t(\theta) = \sum_{a=0}^{2l} A_j^a[t](\theta - \theta_j)^a$. The $A_j^a[t]$ coefficients are linear in $t(\theta)$ and are solution of

$$\sum_{a=0}^{2l} A_i^a[t](b\Delta)^a = t(\theta_{i+b}), \qquad b \in \{-l, -l+1, ..., l-1, l\} \,. \tag{105}$$

The last step in the discretization procedure estimates $t(\theta) = \partial_\theta \chi(\theta)$ through a finite difference scheme of the $2d^{\text{th}}$ order, obtained solving for the first derivative the following system

$$\chi(\theta_{i+a}) = \sum_{s=0}^{2d} \frac{(a\Delta)^s}{s!} \partial_\theta^a \chi \Big|_{\theta=\theta_i}, \qquad a \in \{-d, ..., d\} \,. \tag{106}$$

Combining together eq. (104-105-106) we obtain the desired discretization of the TBA equation (102)

$$e^{\chi(\theta_i)} - 1 + f(\theta_i) + \frac{1}{4\beta \cosh \theta_i} \sum_{j=-N+1}^{N} \omega_{i,j} \chi(\theta_j) = 0 \,. \tag{107}$$

The coefficients $f(\theta_i)$ and the matrix $\omega_{i,j}$ are numerically computed once and for all evaluating the needed integrals, then the non-linear equation obtained this way is solved for the discrete set $\chi(\theta_i)$ through the iterative Newton method. The algorithm is observed to have fast convergence properties using as starting point the non-interacting solution, i.e. $\chi = 0$. For $\beta \sim \mathcal{O}(1)$, with $N = 800$, $\Delta = 0.0175$ and $a = d = 3$, the TBA equation (102) is solved within an error $\sim 10^{-8}$. Once the needed TBA solutions are obtained the remaining passages follow smoothly: the linear dressing equations, which involve the same singular kernel of the TBA, are discretized again following eq. (104-105-106) and then solved through a matrix inversion. The nonlinear equation defining the solution of the partitioning protocol (23) is solved by simple iteration, displaying fast convergence.

## C.2 Direct numerical simulation

The formal definition of the averages we want to compute is given in (1). We compute numerical values for these averages by the Metropolis-Hasting algorithm applied to a discretisation of the Sinh-Gordon field theory on a lattice. This algorithm generates a sequence of $N$ field configurations $\mathscr{C}_i$ (at time $t = 0$) such that the statistical average is approximated by the average of the values on the configurations

$$\langle \mathcal{O} \rangle \simeq \frac{1}{N} \sum_{\mathscr{C}_i} \mathcal{O}(\mathscr{C}_i) \,, \tag{108}$$

where we use the equations of motion to find the value of $\Phi(x, t)$ from the initial configuration $\mathscr{C}$, if needed to calculate the value of $\mathcal{O}$.

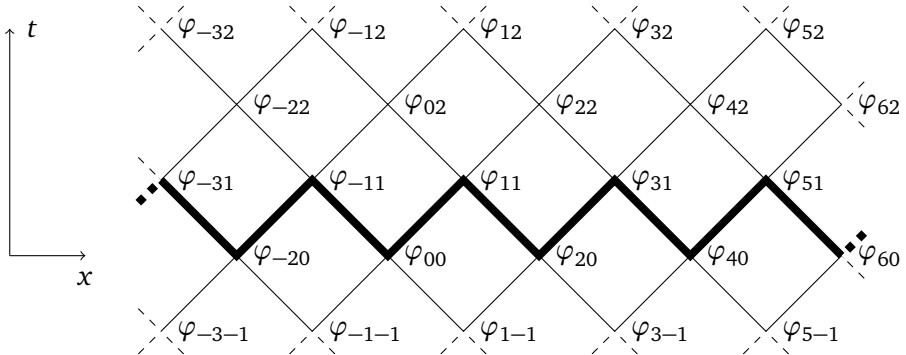

Figure 10: The light-cone lattice and the field $\varphi_{ij}$

For various reasons (outlined below) we have chosen an integrable discretisation of the Sinh-Gordon field theory on a light-cone lattice. We have used the description of Hirota's integrable discretization of the Sine-Gordon model given by Orfanidis in [82] and analytically continued this to obtain the discrete Sinh-Gordon model.

The discrete model lives on a light-cone lattice, as shown in figure 10. We use the variable $\varphi = \exp(g\Phi/2)$, and denote the lattice variables by $\varphi_{ij} \equiv \varphi(ai, aj)$ where $a$ is the lattice spacing as this simplifies the evolution equations and conserved quantities.

The time evolution is

$$\varphi_{i,j+1} = \frac{1}{\varphi_{i,j-1}} \left( \frac{\varphi_{i-1,j}\varphi_{i+1,j} + \lambda}{1 + \lambda\varphi_{i-1,j}\varphi_{i+1,j}} \right) , \quad \lambda = \frac{a^2 m^2}{4} . \tag{109}$$

We fully specify the field at each lattice point if we give the values $\{\varphi_{2i,0}, \varphi_{2i+1,1}\}$ on the thick zig-zag line in Figure 10, so a configuration $\mathscr{C}$ for us is the set of values $\{\varphi_{2i,0}, \varphi_{2i+1,1}\}$ and this is the data used in the Metropolis algorithm to find a thermal or GGE ensemble.

One starts by assigning values randomly to an initial configuration $\{\varphi_{2i,0}, \varphi_{2i+1,1}\}_0$ Given a configuration $\{\varphi_{2i,0}, \varphi_{2i+1,1}\}_n$ (with the values assigned randomly) we consider a new configuration obtained by picking a site, picking a new value of the field at that site (through a suitable random process) and accepting or rejecting the new value according to the Metropolis algorithm. The resulting set of values is the new configuration $\{\varphi_{2i,0}, \varphi_{2i+1,1}\}_{n+1}$. This is then repeated, running through all the sites in the lattice in turn. One has to tune the process of choosing a new field value to maximise the rate at which the sequence of field configurations sample the total configuration space. The successive field configurations will be correlated, so that one has to discard an initial number which are correlated with the initial random configuration; after that, one collects a total number $N$ of samples, with $N$ large enough for the numerical approximation (108) to converge. The details of the Metropolis-Hasting algorithm are left to the original references [80, 81].

There were several reasons for choosing this integrable discretisation, the principle one being that for many quantities we need to evolve the field configurations in time. We found that the errors introduced by a non-integrable discretisation of the Sinh-Gordon model required a very small time step in the numerical time evolution, much smaller than the spatial separation in the discretised model as was also found in [69]. With an integrable discretisation we could use the same size step in the two directions (technically in both light-cone directions) which meant that the time-evolution was not only up to 100 times faster but had intrinsically smaller errors. Another advantage of the lightcone discretisation is that one only needs to store the values of the field $\Phi$ at each site, and not the pair $(\Phi, \Pi)$, the reason being that the conjugate momentum to $\Phi$ on the lightcone is the lightcone derivative of $\Phi$ itself and not an independent field.

These evolution equations are integrable with an infinite set of conserved quantities as shown in [82], the simplest of which are the light-cone components of the momentum,

$$\mathscr{P}_{\pm} = (2a) \sum_{i \in 2\mathbb{Z}} \mathscr{P}_{\pm,i} , \tag{110}$$

$$\mathscr{P}_{\pm,i} = \frac{1}{a^2 g^2} \left( \frac{\varphi_{i,0}}{\varphi_{i\pm1,1}} + \frac{\varphi_{i\pm1,1}}{\varphi_{i,0}} - 2 \right) + \frac{\lambda}{a^2 g^2} \left( \varphi_{i,0} \varphi_{i\mp1,1} + \frac{1}{\varphi_{i\mp1,1} \varphi_{i,0}} - 2 \right) . \tag{111}$$

The energy and momentum are given by

$$H = \mathscr{P}_+ + \mathscr{P}_- , \;\; P = \mathscr{P}_+ - \mathscr{P}_- . \tag{112}$$

The next simplest conserved charges $Q_{\pm3}$ have spin $\pm3$, which are given in the continuum as space integrals of two conserved currents. These can easily be constructed explicitly in light-cone coordinates $x^{\pm} = t \pm x$ as, for example,

$$Q_{\pm3} = \int (T_{\pm4} + T_{\pm2}) \mathrm{d}x , \;\; \partial_{\pm} T_{\pm2} + \partial_{\mp} T_{\pm4} = 0 , \tag{113}$$

$$T_{\pm4} = (\partial_{\pm}^2 \Phi)^2 + \frac{g^2}{4} (\partial_{\pm} \Phi)^4 , \;\; T_{\pm2} = \frac{m^2}{4} (\partial_{\pm} \Phi)^2 \cosh(g\Phi). \tag{114}$$

Note that these expressions are only defined up to the addition of total derivatives.

One can likewise define exactly conserved lattice conserved charges which have these as their continuum limit. There is a construction given in [82], but we will instead use the identification (after a charge of variables) of the continuation of Orfanidis' equations with the integrable equations of type $H3_{\delta=0}$ in the classification of Adler et al. [83] and use the simpler expression given by Rasin and Hydon in table 2 of [84]. After the required change of variables and using the evolution equation to express the conserved quantities (originally defined on a light cone) in terms of the field values on the initial zig-zag line we find

$$\mathscr{Q}_{\pm3} = (2a) \sum_{i \in 2\mathbb{Z}} \mathscr{Q}_{\pm3,i} \tag{115}$$

$$\mathscr{Q}_{\pm3,i} = \frac{1}{a^4 g^2} \Big( \log[\varphi_{i\pm2,0} \varphi_{i+1,1} \varphi_{i,0} \varphi_{i-1,1}] + 2\log[2(1+\lambda)]$$

$$- 2\log\big[\varphi_{i\pm2,0} \varphi_{i,0} + \varphi_{i+1,1} \varphi_{i-1,1} + \lambda(1 + \varphi_{i\pm2,0} \varphi_{i,0} \varphi_{i+1,1} \varphi_{i-1,1})\big]\Big). \tag{116}$$

There are both conserved by the evolution equation but are not finite in the $a \to 0$ limit. The simplest combination which gives the expected form $m^3 \cosh(3\theta)$ in the continuum limit is

$$H_3 = 4\left(\mathscr{Q}_{+3} + \mathscr{Q}_{-3} + \frac{1}{a^2} H\right). \tag{117}$$

The thermal and GGE ensembles are then constructed using the expressions (112) and (117) for $H$ and $H_3$. For the homogeneous case, the lattice Hamiltonian was simply given by

$$\beta_1 H_1 + \beta_3 H_3 , \tag{118}$$

and we chose periodic boundary conditions for the lattice variables, so that if there are $n$ sites then $\varphi_{1j} = \varphi_{n+1j}$. For the partitioning protocol chose a periodic lattice of length $2n$ where the lattice Hamiltonian was

$$\sum_{i=1}^{n} (\beta_1^L \cdot (\mathscr{P}_{+i} + \mathscr{P}_{-i}) + \beta_3^L \cdot 4(\mathscr{Q}_{+3i} + \mathscr{Q}_{-3i} + \frac{1}{a^2}(\mathscr{P}_{+i} + \mathscr{P}_{-i}))) \tag{119}$$

$$+ \sum_{i=n+1}^{2n} (\beta_1^R \cdot (\mathscr{P}_{+i} + \mathscr{P}_{-i}) + \beta_3^R \cdot 4(\mathscr{Q}_{+3i} + \mathscr{Q}_{-3i} + \frac{1}{a^2}(\mathscr{P}_{+i} + \mathscr{P}_{-i}))) . \tag{120}$$

This meant that the two halves of the periodic lattice were joined while in equilibrium in their separate GGE "heat baths" at $t = 0$, before being allowed to evolve freely for $t > 0$. This meant that there was a small, smooth, transition zone around the junctions of the two separate baths, but this does not seem to have had a big effect on the long time behaviour. We had considered preparing the two halves each in their own heat bath and then joining them at $t = 0$, but the discontinuity in the fields and their derivatives created a "spike" of energy at the junction which then persisted for long times.

It is straightforward to include higher charges in the lattice Hamiltonian. The lattice equations of motion (in their H3$_{\delta=0}$ formulation) have been investigated extensively and several constructions of infinite series of conserved quantities are known, see [85,86]. The terms in the conserved charges depend on increasing numbers of lattice variables: $H$ depends on only two nearest neighbour sites, $H_3$ on four neighbouring sites, and $H_{2n+1}$ on $2n+2$ neighbouring sites. In terms of the numerical evaluation, this means a small increase in the computational time to evaluate the Hamiltonian but the main effect is the suppression of spatial variation through the coupling to higher powers of the derivative which mean that the standard Metropolis algorithm takes much longer to converge. Serious numerical work including $H_5$ or higher might need a more sophisticated algorithm for updating the configurations than the simple one outlined here.

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
