# Peer review of "Generalized hydrodynamics of classical integrable field theory: the sinh-Gordon model"

_SciPost Physics, doi:SciPost Phys. 4, 045 (2018)_

## Round 2 · Referee Report · Anonymous (Referee 1) · 2018-3-5

Strengths

  1. The subject considered is timely and interesting.
  2. The problem is original and relevant to the development of further research.
  3. The results are solid and clearly stated.

Weaknesses

  1. I find that some notions are not sufficiently defined/explained.
  2. In some aspects, I think the clarity of presentation should be improved.

See report for details.

Report

The paper applies the newly introduced formalism of Generalised Hydrodynamics (GHD) to classical field theory, more specifically the sinh-Gordon model. The GHD itself is partly based on conjectures, so besides extending the formalism to a new class of models, this work also provides an important test of its validity. In addition, the results also provide a test of the validity of a recursive formula for the one-point function of exponential operators. These results warrant the publication of this paper as Tier II, after the authors considered the suggestions for improvement listed below.

Requested changes

  1. The authors use the notion "Euler scale", but it's not defined anywhere. Presumably this is meant to ebe the scale on which macroscopic hydrodynamics is valid, but this is unclear. Also, is it possible to give some more specific characterisation of this regime (e.g. some minimum length scale specified in terms of the microscopic dynamics)?
  2. What do the authors mean when writing "Euler-scale correlation functions are more precisely obtained by averaging over fluid cells"? Is this a more detailed and precise definition of the correlation functions, or it is the case that numerical precision is enhanced by the averaging?
  3. In the paragraph after eqn. (36) the authors specify that the starting configurations involve only chemical potentials coupled to the charge Q1, or to Q1 and Q3. This is however not sufficiently emphasized and the use of continuous text with inline formulae make the passage less readable. I would also like to ask whether in the GGE case inclusion of unequal beta1 on the two sides, and/or a further charge (say Q5) would present much difficulty. I think that including such cases would make the results substantially more convincing and general.
  4. With respect to Figure 3, could the authors provide a quantitative measure for the deviation from the free result to demonstrate that the interacting case provides a better fit (a possibility would be the integrated deviation ratio they use to quantify the matching between the numerical simulation and the interacting GHD). Alternatively, is there a choice for the parameters beta and g such that the difference is more observable then for the case they plot in Figure 3?
  5. In the first paragraph of Section 3.3, I find that the use of the inline formulae make the text much less readable (maybe this is subjective, though).

---

## Round 2 · Referee Report · Anonymous (Referee 2) · 2018-3-9

Strengths

  1. The subject of the paper is currently in the focus of research.
  2. New analytic results are derived for the classical sinh-Gordon model.
  3. The paper provides direct evidence for a number of conjectured expressions and methods (see Report).

Weaknesses

  1. The clarity of the presentation could be slightly improved.

Report

The so-called Generalised Hydrodynamics (GHD) is a young theory that describes large (hydrodynamic, "Eulerian") scale behaviour of integrable systems, originally developed in the quantum case. The paper contributes to this fresh field of research by generalising GHD to classical field theories, which offers a way of testing some of its ideas and explicit expressions. Comparing the predictions with independent numerical simulations is easier in the classical case (especially in the continuum). The comparison provides evidence that the GHD formalism is valid even for classical field theories having radiative modes, which is an interesting new result.

In particular, the numerical tests verify in the classical domain - the GHD formula for correlation functions, - the so-called hydrodynamic projection method that allows for the computation of correlations of operators that are not conserved densities or associated currents, - a recently proposed expression for GGE expectation values of vertex operators.

Based on this, I recommend publication of the manuscript.

I have the following suggestions and questions to the authors which in my opinion could further improve the paper:

  1. Perhaps the physical situations considered in Sec. 3.2 and Sec. 3.3 could be stated more clearly and explicitly, even in the Figure captions.

  2. The authors may want to consider extending the content of Sec. 3.3 to the case of a GGE state or even to the partitioning protocol.

  3. Can the V-functions of vertex operators be derived from the semiclassical limit of form sinh-Gordon form factors?

  4. The method for the initial state preparation is not given, only a reference is cited. I think it deserves a couple of sentences at least in the Appendix.

  5. It is not immediately clear why an integrable discretisation of the field equations leads to smaller statistical errors.

Some typos:

  • in the 3rd paragraph of the Introduction the parenthesis of "(see also" is not closed.

  • it seems that a factor of g is missing from the argument of cosh in Eqs. (48) and (49).

  • the punctuation of the sentence around Eqs. (52) and (53) is not clear.

Requested changes

  1. Clearer motivation and presentation of the setups studied in Sec. 3.2 and 3.3.

  2. Some details on the initial state preparation.

---

## Round 3 · Referee Report · Anonymous · 2018-5-16

Report

I am satisfied with the changes the authors made to the text, and their answers, and so I recommend the paper for publication in SciPost Physics.

---

## Round 3 · Referee Report · Anonymous · 2018-6-13

Report

The authors answered my questions and took my suggestions into account which improved the clarity of the presentation. I recommend the manuscript for publication.

---

## Round 3 · Author Response

In the following, we respond to the referee reports and present a list of changes in the revised version of our manuscript.

REFEREE 1:

We thank the Referee 1 for having recommended our paper for publication in SciPost and for his/her comments, which we address hereafter

-Referee: The authors use the notion "Euler scale", but it's not defined anywhere. Presumably this is meant to be the scale on which macroscopic hydrodynamics is valid, but this is unclear. Also, is it possible to give some more specific characterization of this regime (e.g. some minimum length scale specified in terms of the microscopic dynamics)?

Answer: The "Euler scale" is a concept that is used in standard hydrodynamic literature, see for instance [6] (here and below, new numbering is used). It refers to the scaling limit whereby parameters characterizing the state are taken to vary in space on a large scale $\ell$, observables are taken at large space and time also proportional to $\ell$ and averaged over fluid cells, correlations are scaled appropriately in $\lambda$, and the limit $\ell\to\infty$ is taken. A precise description can be found in quite some generality for instance in [18,25]. We have clarified this in the third paragraph of the introduction. Note that in the present paper, we provide both a precise definition of Euler-scale correlations in GGEs, see (24, 28), and numerical evidence for some of the details (in particular, the fluid cell averaging). In the partitioning protocol, a lot of evidence for emergence of hydro exists, e.g., besides the present work, in Physical review letters 117 (20), 207201, Physical Review B 96 (11), 115124 and Journal of Statistical Mechanics 2017, 073210, but no precise result concerning the space/time scales actually needed is available yet. In fact, apart from free theories, a derivation of GHD is still out of hand.

-Referee: What do the authors mean when writing "Euler-scale correlation functions are more precisely obtained by averaging over fluid cells"? Is this a more detailed and precise definition of the correlation functions, or it is the case that numerical precision is enhanced by the averaging?

Answer: This meant a more detailed and precise definition of the Euler-scale correlation functions. We have taken away the ambiguous "more precisely", added a reference to [25] for a full definition, and mentioned that we numerically verify and analytically test the necessity of fluid cell averaging.

-Referee: In the paragraph after eqn. (36) the authors specify that the starting configurations involve only chemical potentials coupled to the charge Q1, or to Q1 and Q3. This is however not sufficiently emphasized and the use of continuous text with inline formulae make the passage less readable. I would also like to ask whether in the GGE case inclusion of unequal beta1 on the two sides, and/or a further charge (say Q5) would present much difficulty. I think that including such cases would make the results substantially more convincing and general.

Answer: We replaced the inline sentences with a more schematic presentation, enhancing the readability of the text, and we largely extended the discussion of the precise protocol and initial state taken. Concerning the suggestion of using different parameters in the partitioning protocol: in the case where Q3 was present it was our intention not only to enhance the role of the latter, but also to be able to access non-equilibrium currents, which are UV divergent in a protocol starting from purely thermal states. Taking unequal $\beta_1$ on both sides is indeed possible - we have added a study of this. Concerning Q5: while in principle numerically feasible, the derivation of the explicit expression of Q5 from the integrable discretization (and its matching with the continuous case) is expected to be quite complicated, and this is not expected to provide much new physics (we already have the non-equilibrium current with Q3).

-Referee: With respect to Figure 3, could the authors provide a quantitative measure for the deviation from the free result to demonstrate that the interacting case provides a better fit (a possibility would be the integrated deviation ratio they use to quantify the matching between the numerical simulation and the interacting GHD). Alternatively, is there a choice for the parameters beta and g such that the difference is more observable then for the case they plot in Figure 3?

Answer: We included the required measure in the text. In particular, the distance between the interacting/free curve, when compared with the distance interacting/numerics, is larger of an order of magnitude. This is clearly displayed in Figure 3: the black and red lines are exactly one on the top of the other, while the dashed green curve clearly departs from both. We feel like the precision of the numerics is enough to tell apart GHD and free field result, deciding in favor of the first. We also added an extra comparison graph, in order to fully discard the possibility that our data may be obtained by a free-theory calculation, by choosing temperatures in the baths such that the trace of the stress-energy tensor in the free case would match those of the interacting case. We observe, in the profile between the baths, a measurable difference, beyond the imprecision of our numerics. Finally, as depicted in Figure 1, the difference between the expectation values of the stress energy tensor in the free and interacting case is enhanced increasing the coupling or, equivalently, the temperature. However, for the numerics, exploring larger temperatures results in a worse approximation of the continuum by mean of the discretization. This would then require a smaller lattice step, slowing down the algorithm and ultimately resulting in larger statistical errors. Hence we have opted to avoid this and keep the parameters we have.

-Referee: In the first paragraph of Section 3.3, I find that the use of the inline formulae make the text much less readable (maybe this is subjective, though).

Answer: We tried to modify the text promote the definition of the needed quantities from inline expressions to standard equations, however this resulted in several breaks in the text flow. In the end, we feel like the previous disposition is more readable and thus preserved it.

REFEREE 2:

We thank Referee 2 for his/her positive opinion on our work and for recommending its publication on SciPost. We modified the manuscript accordingly to his/her suggestions. We thank the referee for having pointed out a few typos that we corrected. Hereafter, we address one by one the questions of the referee

-Referee: Perhaps the physical situations considered in Sec. 3.2 and Sec. 3.3 could be stated more clearly and explicitly, even in the Figure captions.

Answer: At the beginning of Sec. 3.2 we further clarified the reasons for choosing the partitioning protocol, and we provided much more detail on the explicit protocol used in our numerics. For what it concerns Sec. 3.3, we attempted to clarify that we consider correlations in homogeneous thermal ensembles. We think that looking at correlation functions in the thermal ensemble is probably the most natural question to be asked in equilibrium statistical physics, and we emphasize that the exact GHD expressions that we check, for dynamical correlation functions even in thermal ensembles, have never been checked before. Note also that the correlation functions involve, in any state (GGE or not), all conserved charges, as Euler-scale correlations are due to the ballistic transport of all conserved quantities that connects the observables, independently of how many conserved quantities are involved in the state itself.

-Referee: The authors may want to consider extending the content of Sec. 3.3 to the case of a GGE state or even to the partitioning protocol.

Answer: This is indeed a good question and suggestion along the lines of arXiv:1711.04568, where correlation functions in inhomogeneous states in the (quantum) Sinh-Gordon have been derived, also inspired by the present work. However, we feel like the thermal case is an enough convincing benchmark. In particular, correlations in the partitioning protocol would require quite additional explanations, especially with respect to the analytical expressions themselves, which would require somewhat involved numerical analysis. It must also be stressed that, in order to reach the high numerical precision required to test the correlation functions, a single simulation takes several days. The situation would be even worse for the partitioning protocol, where we cannot take advantage of translational invariance. As for GGE states, these are not expected to provide big differences with respect to thermal states. Again, as stated above, thermal state already provide checks of the integrability structure behind Euler scale correlations, as they are carried by all conserved quantities.

-Referee: Can the V-functions of vertex operators be derived from the semiclassical limit of form sinh-Gordon form factors?

Answer: The original proposal to consider expectation values of local observables was indeed through the classical limit of the Le Clair-Mussardo expansion, namely a form factor expansion [A. De Luca and G. Mussardo J. Stat. Mech. (2016) 064011].
From such an expansion it can be in principle derived a form factor expansion for the V functions of our work. However, such an expansion (if truncated) is necessarily a low energy expansion, thus not suitable to highly excited thermal states such those we considered. Our result derived from the Smirnov-Negro formula is equivalent to the (extremely challenging) resummation of the form factor expansion.

-Referee:The method for the initial state preparation is not given, only a reference is cited. I think it deserves a couple of sentences at least in the Appendix.

Answer: We added a paragraph in the Appendix sketching the idea behind the Metropolis algorithm.

Referee: It is not immediately clear why an integrable discretisation of the field equations leads to smaller statistical errors.

Answer: The integrable discretization is not needed to cope with statistical errors, but rather with those introduced by time evolution. These may introduce thermalizing effects which might make the verification of GHD predictions (based on integrability) more tricky. See "In particular, an arbitrary discretization will, in general, spoil the integrability of the model. This will not only affect the dynamics, but will introduce thermalizing effects which may make the verification of the GHD formulae, strongly based on integrability, more tricky." at the beginning of C.2. A discretized, approximate time evolution breaking integrability would requrie rather small time discretization in order to approximately conserve all the charges. This would have ultimately heavily slowed down each single-field time evolution making difficult to accumulate the needed number of samples in a sustainable amount of time.

---

## Editorial Decision

published